# The Role of NK Cells in Cancer Immunotherapy: Mechanisms, Evasion Strategies, and Therapeutic Advances

**DOI:** 10.3390/biomedicines13040857

**Published:** 2025-04-02

**Authors:** Paula Morcillo-Martín-Romo, Javier Valverde-Pozo, María Ortiz-Bueno, Maurizio Arnone, Laura Espinar-Barranco, Celia Espinar-Barranco, María Eugenia García-Rubiño

**Affiliations:** 1Centre for Biomedical Research (CIBM), University of Granada, 18100 Granada, Spain; e.pmormartin@go.ugr.es (P.M.-M.-R.); mauriarnone@correo.ugr.es (M.A.); 2Departamento de Química y Bioquímica, Facultad de Farmacia, Universidad San Pablo-CEU, CEU Universities, Urbanización Montepríncipe, 28668 Boadilla del Monte, Spain; javier.valverdepozo@ceu.es; 3Nanoscopy-UGR Laboratory, Department of Physical Chemistry, Faculty of Pharmacy, Unidad de Excelencia en Quimica Aplicada a Biomedicina y Medioambiente (UEQ), University of Granada, C. U. Cartuja, 18071 Granada, Spain; mariaortbue@correo.ugr.es (M.O.-B.); laura.espinarbarranco@mssm.edu (L.E.-B.); 4Department of Medicine, Translational Transplant Research Center, Immunology Institute, Icahn School of Medicine at Mount Sinai, New York, NY 10029, USA; 5Servicio de Análisis Clínicos e Inmunología, Hospital Universitario Virgen de las Nieves, 18014 Granada, Spain; celia.espinar.sspa@juntadeandalucia.es; 6Instituto de Investigación Biosanitaria ibs.GRANADA, University Hospitals of Granada-University of Granada, 18100 Granada, Spain

**Keywords:** natural killer cells, cancer immunotherapy, tumor evasion, CAR-NK cells, immune checkpoint inhibitors, tumor microenvironment, adoptive cell therapy, cytotoxicity, innate immunity, cytokine-based NK cell therapy

## Abstract

**Background/Objectives:** Natural killer (NK) cells play a crucial role in tumor surveillance by exerting cytotoxic activity and modulating immune responses. However, tumors employ diverse evasion strategies that limit NK cell effectiveness. This review aims to explore the molecular mechanisms of NK cell activation and inhibition in cancer, the influence of the tumor microenvironment, and the latest advancements in NK cell-based immunotherapies, including adoptive NK cell transfer and Chimeric Antigen Receptor-Natural Killer (CAR-NK) cell therapies. **Methods:** A comprehensive literature review was conducted, prioritizing peer-reviewed studies from the last decade on NK cell biology, tumor immune evasion, and immunotherapeutic applications. The analysis includes data from preclinical models and clinical trials evaluating NK cell expansion strategies, cytokine-based stimulation, and CAR-NK cell therapy developments. **Results:** NK cells eliminate tumors through cytotoxic granule release, death receptor pathways, and cytokine secretion. However, tumor cells evade NK-mediated immunity by downregulating activating ligands, secreting immunosuppressive molecules, and altering the tumor microenvironment. Novel NK cell-based therapies, such as CAR-NK cells and combination approaches with immune checkpoint inhibitors, enhance NK cell persistence and therapeutic efficacy against both hematologic and solid malignancies. Clinical trials suggest improved safety profiles compared to CAR-T therapies, with reduced cytokine release syndrome and graft-versus-host disease. **Conclusions:** While NK cell-based immunotherapies hold great promise, challenges remain, including limited persistence and tumor-induced immunosuppression. Addressing these hurdles will be critical for optimizing NK cell therapies and advancing next-generation, off-the-shelf immunotherapeutics for broader clinical applications.

## 1. Introduction

Natural killer (NK) cells are a fundamental component of the innate immune system, playing a crucial role in immune surveillance and tumor eradication [1]. Unlike T cells, NK cells do not require prior antigen sensitization, enabling them to rapidly recognize and eliminate malignant cells through cytotoxic mechanisms and immune modulation [2]. Their ability to distinguish between healthy and transformed cells is governed by a delicate balance between activating and inhibitory signals [3].

Despite their intrinsic antitumor properties, tumor cells have evolved sophisticated evasion strategies to suppress NK cell activity. These mechanisms include the downregulation of activating ligands, overexpression of inhibitory molecules, and the creation of an immunosuppressive tumor microenvironment [4,5,6]. As a result, NK cell function is often impaired in patients with cancer, limiting their effectiveness as a natural defense mechanism [7].

In recent years, NK cell-based immunotherapies have gained significant attention as a promising alternative to T cell-based approaches, such as CAR-T cell therapy. Strategies such as adoptive NK cell transfer, cytokine-based activation, and genetically engineered CAR-NK cells have shown encouraging results in preclinical and clinical studies [8,9]. Notably, CAR-NK cells exhibit a favorable safety profile compared to CAR-T cells, with reduced risks of cytokine release syndrome and graft-versus-host disease [10,11]. This review aims to provide an in-depth analysis of NK cell activation and inhibition mechanisms in cancer, the impact of the tumor microenvironment on NK cell function, and the latest advancements in NK cell-based immunotherapies [12]. Specifically, we integrate findings from foundational works such as those by Nutt and Huntington on NK cell biology, Vivier et al. [2] on NK cell-based therapies, and Buckle and Guillerey on the checkpoint receptor T cell immunoreceptor with Ig and ITIM domains (TIGITs), and studies on the interaction of nectin-1 with CD96 will be examined to highlight critical regulatory pathways [1,2,3].

Several comprehensive reviews have been published over the past few years, addressing various aspects of NK cell biology and their therapeutic potential in cancer immunotherapy. For instance, the review “Understanding NK cell biology for harnessing NK cell therapies: targeting cancer and beyond” by Shin et al. offers a thorough overview of NK cell development, functional mechanisms, and therapeutic applications, including CAR-NK and Antibody-Dependent Cellular Cytotoxicity (ADCC)-enhancing strategies [13]. Similarly, “Allogeneic natural killer cell therapy” by Berrien-Elliott et al. discusses NK cell manufacturing, persistence, and engineering strategies to overcome tumor immune evasion and to develop off-the-shelf products [14]. The review “CAR-T and CAR-NK as cellular cancer immunotherapy for solid tumors” by Peng et al. highlights the challenges specific to solid tumors and compares the advantages and limitations of CAR-NK over CAR-T cells [15].

Despite these advances, there are still areas that require further clarification. These include improving the persistence and in vivo expansion of NK cells, overcoming the immunosuppressive tumor microenvironment, and optimizing genetic engineering techniques for safer and more effective CAR-NK therapies. Moreover, the translation of promising preclinical strategies into effective treatments for solid tumors remains a major hurdle.

This review will also discuss the current challenges in the field, including issues related to persistence, expansion, and tumor immune evasion [16]. Understanding these aspects will be crucial for optimizing NK cell therapies and expanding their clinical applications [17].

By integrating both foundational immunological mechanisms and the most recent therapeutic developments, this review offers a comprehensive and updated perspective that complements the existing literature and highlights emerging strategies to overcome current barriers in NK cell-based cancer therapy.

## 2. Mechanisms of NK Cell Activity in Cancer

### 2.1. Recognition and Signaling Pathways

NK cells are innate cytotoxic lymphocytes that can lyse cancer or virus-infected cells. Their activity is regulated by a wide variety of receptors that can mediate activating or inhibitory signals. They can exert their cytotoxic activity through two distinct pathways; by releasing cytotoxic granules containing perforin and granzymes, or by inducing apoptosis mediated by receptors expressed on tumor cells. The ability of NK cells to identify and eliminate tumor cells depends on a dynamic balance between activating and inhibitory signals. This section examines the key receptors involved in this process and their implications in immune regulation and tumor evasion.

#### 2.1.1. Recognition of Tumor Cells by Inhibitory and Activating Receptors on NK Cells

NK cells are fundamental components of the innate immune system, specialized in the elimination of tumor cells and cells infected by pathogens. Their cytotoxic capacity is regulated by inhibitory and activating receptors, which allow NK cells to discriminate between normal cells and those with molecular alterations, such as the loss of MHC class I or ligand expression in situations of cellular stress, as is the case in tumor cells [1,2].

Inhibitory receptors on NK cells play a crucial role in tolerance and self-recognition toward self-cells by recognising the MHC-I molecules present in healthy cells. In humans, the main inhibitory receptors belong to the killer immunoglobulin-like receptors (*KIRs*) family and to the *CD94/NKG2A* complex. In rodents, this function is predominantly performed by the Ly49 family. These receptors contain immunoreceptor tyrosine-based inhibition motifs (ITIMs) in their cytoplasmic domains, which, when activated by binding to MHC class I, recruit phosphatases such as SHP-1 and SHP-2. These phosphatases attenuate activating signals by dephosphorylating key components of intracellular signaling pathways [1,2,3,4].

NK cells have the ability to recognize and eliminate those cells that have lost or decreased MHC class I expression, frequent in tumor cells and those infected by viruses, since the absence of MHC class I prevents inhibitory receptor-mediated inhibition. In addition, it has been observed that in tumor cells, the increased expression of receptors such as *KIR* and *CD94/NKG2A* is associated with an inhibitory phenotype that reduces NK cell cytotoxicity in the tumor microenvironment [2,18,19,20]. This inhibitory phenotype refers to a functional state in which NK cells exhibit reduced cytotoxicity and cytokine production due to the upregulation of inhibitory receptors—such as *NKG2A*, *TIGIT*, or *PD-1*—and the influence of immunosuppressive signals within the tumor microenvironment. This intricate balance between activation and inhibition ensures that NK cells respond effectively to tumor cells while sparing healthy tissues.

*KIR* receptors belong to a family of 15 genes linked to chromosome 19 with a high degree of polymorphisms, which recognize several HLA isoforms (MHC-I molecules) such as HLA-A, HLA-B, and HLA-C. Crucially, the activation of these receptors inhibits the destruction of normal cells containing these HLA molecules, protecting them from NK cells. In contrast, when tumors or infected cells lose MHC-I expression, KIRs are not activated and NK cells can exert their cytotoxic action [1]. This loss of MHC-I expression by tumor cells not only activates NK cell cytotoxicity but also highlights the intricate mechanisms tumors use to evade immune responses, as discussed further in this section.

The CD94/NKG2 complex is another inhibitory receptor in which a single CD94 gene is linked to four *NKG2* genes (A, C, E, and F), and is present in both rodents and humans. CD94 is found on the cell surface and can be alone or associated with activating (C, E) or inhibitory (A) forms of NKG2. This complex recognizes non-classical MHC-I ligands such as HLA-E [1,19].

Other families of inhibitory receptors have been identified on NK cells, namely nectin receptors (*TIGIT*, CD96, and CD112R), which inhibit NK cell functions by binding to nectin molecules expressed on tumor cells. These receptors bind to CD155 (Nexl-5 or PVR) and CD112 (Nectin-2), two ligands ubiquitously expressed at low levels and often found elevated on tumor cells. In addition, CD155 and CD112 are expressed on myeloid cells within the tumor microenvironment [3,21].

*TIGIT* (T cell immunoreceptor with Ig and ITIM), also called VSig9, Vstm3, or WUCAM, is an inhibitory receptor expressed on T cells and NK cells. Its ligands are CD155 (Nexl-5 or PVR), CD112 (nectin-2), and CD113 (nectin-3). TIGIT competes with *DNAM-1* (activating receptor) to bind CD155 and CD112. In tumor cells, this signaling reduces the cytotoxic activity of NK cells [22,23]. Understanding how TIGIT modulates NK cell activity opens new avenues for immunotherapeutic strategies, particularly in overcoming tumor-induced immune suppression.

In addition to *TIGIT*, other nectin receptors like *CD96* also contribute to the regulation of NK cell activity. CD96, also called TACTILE (T cell-activated increased late expression), is expressed on both mouse and human NK cells in the basal state. CD96 also binds to CD155 and possesses two cytoplasmic domains, one based on tyrosine (ITIM) and the other on a proline-rich sequence [21,23,24].

Further studies are still needed to determine whether the role of CD96 is the activation or inhibition of NK cells. On the one hand, it has been shown that the cytotoxicity of human NK cell lines increased in the presence of CD96; however, in mice, there is evidence that CD96 inhibits the antitumor activity of NK cells, mainly by decreasing IFNγ production [6,24].

Together, *CD96*, DNAM-1, and *TIGIT* form a regulatory system that controls the activation or inhibition of NK cells by interacting with CD155, nectin-2, and nectin-1 [6,21,23]. To provide a concise visual summary, Table 1 outlines the main NK cell receptors, classifying them as activating or inhibitory, along with their respective ligands, signaling pathways, and immune functions.

CD112R (or PVRIG) is a receptor that binds CD112 with much higher affinity than *DNAM-1* or *TIGIT*. The expression of this receptor on NK cells has been correlated with increased expression of inhibitory receptors and reduced cytotoxicity, and its blockade with increased antitumor response [3,21,25].

In addition to these receptors, we found T cell-associated checkpoints that are also expressed on NK cells, especially under conditions of chronic activation or tumors such as the following: lymphocyte activation gene 3 (LAG-3), T cell immunoglobulin and mucin domain 3 (TIM-3), *PD-1*, and *CTLA-4.*

*LAG-3* (CD223) was identified as a cell surface molecule expressed on activated T lymphocytes and NK cells, although it was later observed that it is also found on B lymphocytes, regulatory T lymphocytes, and antigen presenting cells (APCs). LAG-3 shares similarities with the CD4 molecule and has the ability to bind to MHC-II molecules more avidly than CD4, thus acting as a CD4 competitor. Although the exact mechanism of LAG-3 remains unclear, evidence suggests its upregulation in chronically activated NK cells. This leads to reduced proinflammatory cytokine production, effectively dampening NK cell cytotoxicity within tumor microenvironments [2,26,27].

TIM-3 is expressed on various cells of the immune system (T cells, NK cells, regulatory T cells, dendritic cells, and macrophages), and some studies suggest that it may be a marker of NK cell activation and differentiation, acquired during maturation at the same time as the acquisition of other inhibitory receptors such as KIR. *TIM-3* marks mature, functional NK cells and restricts their cytotoxic potential in tumor microenvironments, sometimes inducing NK cell apoptosis [2,3,26,27,28,29]. These findings position *TIM-3* as a potential target for enhancing NK cell activity in the tumor microenvironment.

PD-1 (*CD279*) is a programmed cell death receptor that is also expressed on differentiated NK cells. Recent studies have shown that this receptor is associated with reduced degranulation and cytokine production by NK cells in contact with tumor cells, allowing immune evasion of some tumors. Targeting inhibitory receptors like *TIGIT* and *PD-1* holds great potential for enhancing NK cell activity and counteracting tumor-induced immune evasion [3,30,31]. In contrast to inhibitory pathways, NK cell activation is mediated by a diverse set of activating receptors, which recognize stress-induced ligands on target cells and initiate cytotoxic responses.

Cytotoxic T-lymphocyte-associated antigen 4 (*CTLA-4*) is another immune checkpoint expressed on lymphocytes, primarily studied in Tregs and activated effector T cells. While its expression in NK cells remains controversial, some studies suggest that it may be stimulated in the presence of IL-2, potentially inhibiting IFN-γ production [3].

In addition to reducing MHC-I expression, tumor cells can also overexpress ligands for activating receptors on NK cells. These receptors signal through Immunoreceptor Tyrosine-based Activation Motifs (ITAMs) on adaptors like DAP12 or CD3ζ. The phosphorylation of these motifs activates kinases such as Syk and ZAP70, triggering the release of granzymes and perforins to induce apoptosis in target cells [1,32].

Among the main activating receptors are NKG2D (encoded by KLRK1), NKp46, and NKp30 (encoded by NCR3 (CD 337)), which recognize *MICA/B* and *ULBPs*, ecto-calreticulin, and B7-H6, respectively, and are expressed on the surface of stressed cells. Ligands for NKG2D are induced in response to DNA damage recognition, cellular stress, cellular hyperproliferation, p53 activation, and heat shock-induced stress [2,8,9,33,34].

NKp44 (encoded by NCR2) is another activating receptor capable of binding to HLA-DP (MHC-II) on viral peptide presentation, although a tumor ligand has not yet been identified, and the 2B4-CD48 interaction, expressed on NK and tumor cells, respectively, also induces NK cell activation by increasing cytotoxicity and IFNγ secretion [8,34].

Another mechanism for tumor recognition and NK cell activation is mediated by the CD16 receptor (FCGR3A), which binds to the constant region (Fc) of immunoglobulins, called ADCC [2,9].

#### 2.1.2. Mechanisms of Natural Killer Cell-Mediated Cellular Cytotoxicity

NK cells can exert their cytotoxic activity through two distinct pathways. They can release cytotoxic granules containing granzymes and perforins, or induce apoptosis, a programmed cell death through the expression of receptors such as TRAIL and/or Fas ligand (FasL), which bind to TRAIL-R1/-R2 or CD95/Fas, respectively, expressed on tumor or infected cells [10].

Both cytotoxic mechanisms require intimate contact between the NK cell and the target cell. Unlike cytotoxic T cells, which require prior activation and differentiation, NK cells can exert their cytotoxic action directly without prestimulation, although their cytotoxic activity is significantly enhanced in the presence of cytokines and inflammatory signals. NK cells are also capable of modulating the immune response by producing cytokines, especially IFN-γ and TNF, two potent inflammatory cytokines that activate macrophages and lymphocytes at the site of infection or tumor environment. They can release a wide range of cytokines, including the granulocyte macrophage-stimulating factor (GM-CSF), IL-5, IL-10, and IL-13, and chemokines such as CCL3, CCL4, and CCL5 [1,11].

Granzymes are a family of closely related serine proteases that are expressed in cytotoxic T cells and NK cells, and consist of five members: granzyme A, granzyme B, granzyme H, granzyme K, and granzyme M. Together with perforin and granulysin, they represent the main cytotoxic components of the secretory granules of NK cells [10,11,12].

Granulysin, expressed in cytotoxic T cells and NK cells, belongs to the saposin-like protein family. Synthesized in the lysosomal compartment, it is stored in lytic granules. Granulysin exhibits cytolytic activity against tumors and microorganisms. Like other saposin-like proteins, it forms pores in membranes, altering their permeability and causing osmotic lysis and mitochondrial damage [10,16].

Perforin is a protein capable of forming a pore and disrupting the target cell membrane, including the plasma membrane or lysosomal membrane. Once inside the target cell, granzymes are responsible for initiating cell death. Granzymes can function directly by cleaving cellular substrates, such as structural and regulatory proteins found in the nucleus, cytosol, and cytoskeleton [1,10,17], or indirectly by initiating a cascade of proteases. An important substrate of the granzymes is the pro-apoptotic protein Bid (BH3-interacting domain), which moves into the mitochondrion and interacts with the pro-apoptotic proteins Bax and/or Bak. This results in a disruption of the integrity of the mitochondrial membrane, releasing apoptotic factors such as Smac/DIABLO, cytochrome c, apoptosis-inducing factor (AIP), and Omi/HtrA2, leading to cell apoptosis [8,10,11,35].

NK cells employ diverse cytotoxic mechanisms, making them a cornerstone of innate immunity and a promising target for enhancing cancer immunotherapy. This pathway utilizes members of the tumor necrosis factor receptor (TNFR) superfamily that are expressed on target cells. These receptors contain an intracellular motif such as FADD (Fas-associated death domain), which recruits molecules responsible for transmitting the apoptosis signal. The main members of the TNFR family that trigger apoptosis are Fas (CD95) and TRAIL (TNF-related apoptosis-inducing ligand) [1,10].

Fas is expressed in a wide variety of tissues, whereas Fas ligand (FasL) expression is restricted to activated T cells and NK cells, where it is stored in lytic granules and, upon activation, released to the effector cell membrane. After surface expression, it can bind to the CD95 receptor of a target cell and activate its intracellular signaling cascade, which begins with the assembly of the death-inducing signaling complex (DISC) and leads to the activation of a caspase cascade, resulting in the depolarization of the mitochondrial membrane potential and induction of apoptosis [10,36].

TRAIL is a type II transmembrane protein with homology to FasL and TNF. In addition to being expressed in T and NK cells, it can also be found in other cells such as monocytes, dendritic cells, and macrophages. NK cells need to be stimulated with IL-2, IL-15, or IL-12 to express detectable levels of TRAIL on their surface. Human TRAIL can bind to four different membrane receptors and one soluble receptor, although the only ones that can induce apoptosis are TRAIL-1 and TRAIL-2. The proapoptotic signaling of these receptors is very similar to CD95-mediated signaling, with the formation of a DISC and activation of caspase-8 [10,36,37].

By leveraging these cytotoxic pathways, NK cells remain a cornerstone of innate immunity and hold great promise for enhancing immunotherapeutic interventions against cancer. Understanding the complex interplay of activating and inhibitory signals in NK cells provides crucial insights for developing novel immunotherapeutic strategies. Targeting these pathways holds significant promise for improving anti-tumor responses in cancer patients.

#### 2.1.3. Role of Cytokines in the Activation and Expansion of NK Cells in the Tumor Environment

The tumor microenvironment (TME) is a dynamic ecosystem where interactions between tumor cells, immune cells, and cytokines dictate cancer progression or suppression. NK cells, with their innate ability to target stressed or transformed cells, have emerged as an advantageous alternative to T-cell focused therapies. In this review, we explore the release mechanisms of these interleukins within the TME, their triggers, and their roles in NK cell activation and proliferation. Furthermore, we examine how tumors exploit molecular pathways—such as MIC-A/B, MHC-I, and transforming growth factor-beta (TGF-β)—to evade NK cell activity.

IL-2 is a potent T-cell-derived cytokine traditionally associated with the activation and proliferation of T-cells and NK cells. In the TME, IL-2 is mainly secreted by activated CD4^+^ T cells and, to a lesser extent, by dendritic cells and macrophages. IL-2 release is triggered during T-cell activation by antigen presentation by dendritic cells, mediated by MHC-II/TCR interaction and costimulatory signals such as CD28/B7 [38]. Tumors that promote chronic inflammation or recruit activated immune cells may indirectly induce IL-2 production [39].

Upon release, IL-2 binds to the IL-2 receptor (IL-2R) on NK cells, which consists of CD25 (high-affinity α-chain), CD122 (β-chain), and the common γ chain (γc). The binding of IL-2 to IL-2R activates JAK1/3 and STAT5 signaling inside the NK cell, leading to the transcription of genes involved in NK cell proliferation, survival, and production of effector molecules such as granzyme B and perforin. IL-2 also enhances the cytotoxic activity of NK cells and promotes the secretion of IFN-γ, a cytokine that amplifies immune responses and improves antigen presentation by tumor-infiltrating dendritic cells [40].

Despite its potent immunostimulatory effects, IL-2 also has a role in expanding regulatory T cells (Tregs), which suppress NK cell activity within the TME [41].

IL-15 is a critical cytokine for the development, survival, and activation of NK cells. Unlike IL-2, IL-15 is produced constitutively by dendritic cells, macrophages, and stromal cells, and is typically presented in a trans-presentation manner. Stress signals such as hypoxia, DNA damage, and pro-inflammatory cytokines (e.g., IFN-γ and TNF-α) upregulate IL-15 expression in tumor-associated macrophages and dendritic cells [42]. Also, toll-like receptor (TLR) activation on macrophages and dendritic cells by damage-associated molecular patterns (DAMPs) released by dying tumor cells induces IL-15 expression [43].

When released, IL-15 is presented on the cell surface by the IL-15 receptor α (IL-15Rα) of dendritic cells or macrophages in complex with IL-15, engaging the IL-15Rβ/γc heterodimer on NK cells. This interaction activates the JAK1/3 and STAT5 pathways, promoting NK cell survival and proliferation. This interleukin is essential for maintaining NK cell homeostasis and enhances their ability to lyse tumor cells by increasing the expression of activating receptors like NKG2D and NKp46 [44].

Tumors, however, can evade IL-15-mediated NK cell activation by suppressing its expression through factors like IL-10 or by creating an immunosuppressive microenvironment [45].

IL-21 is primarily secreted by activated CD4^+^ T-cells and T-follicular helper (TFH) cells. Its release is triggered during immune activation, particularly in response to antigens presented by dendritic cells and in collaboration with pro-inflammatory cytokines such as IL-12. The chronic inflammation and activation of TFH cells within tertiary lymphoid structures associated with tumors can drive IL-21 production. In addition, tumor antigens processed and presented by dendritic cells can indirectly induce IL-21 secretion by activating T cells [46].

After its release, IL-21 binds to the IL-21 receptor (IL-21R), composed of the IL-21Rα chain and the common γ-chain (γc) shared with IL-2 and IL-15. This binding activates the JAK1/3, STAT3, and PI3K/AKT pathways, which enhances cell cytotoxicity by upregulating the expression of effector molecules like granzyme B and perforin [47]. It also induces FasL expression, enabling NK cells to kill tumor cells through Fas-mediated apoptosis [48].

As shown, NK cells are critical components of the innate immune system, capable of identifying and killing transformed or stressed cells without prior sensitization. Their cytotoxic activity is regulated by a delicate balance of activating and inhibitory signals (Figure 1).

However, tumors develop sophisticated mechanisms to evade NK cell-mediated surveillance, ranging from ligand modulation (e.g., MIC-A/B shedding) to creating an immunosuppressive microenvironment dominated by TGF-β and IL-10, thus contributing to immune escape and tumor progression [49].

### 2.2. Tumor Evasion Mechanisms

#### 2.2.1. Mechanisms of Tumor Escape from Natural Killer Cell-Mediated Immunity

One of the mechanisms by which tumor evades the action of the immune system is the use of the MHC protein complex (Figure 2). MIC-A (MHC class I polypeptide-related sequence A) and MIC-B (MHC class I polypeptide-related sequence B) are stress-induced ligands recognized by the activating receptor NKG2D on NK cells’ surface [50]. These molecules normally are expressed on the surface of stressed or transformed cells, signaling NK cells to eliminate the target, as they are recognized as non-self MHC. Here, we outline some of the mechanisms used by tumors that involve this protein complex, as follows:Proteolytic shedding: Tumor cells release soluble forms of *MICA-B* into the extracellular milieu through proteolytic cleavage mediated by metalloproteinases such as ADAM10 and ADAM17 [51]. The activation of these metalloproteinases is often driven by oncogenic pathways, including the RAS/RAF/MEK/ERK and PI3K/AKT cascades. These pathways increase the transcription and activity of ADAM family proteases, facilitating ligand shedding [52]. Soluble MIC-A/B acts as a decoy, binding to NKG2D on NK cells and internalizing the receptor, thereby impairing NK cell activation [50]. In many tumors that exhibit this mechanism, it often contributes to poor prognosis and malignancy [53].Epigenetic silencing: Tumors downregulate MIC-A/B expression on their surface by modifying their promoters through DNA methylation or histone deacetylation [33]. Hypoxia, a common feature of the tumor microenvironment, induces HIF-1α, which also represses MIC-A/B transcription. HIF-1α directly binds to the promoter regions of MIC-A/B, recruiting co-repressors that inhibit gene transcription [54]. Additionally, the hypoxic environment reduces oxidative stress signals that would otherwise trigger MIC-A/B expression [54].Immune suppression by soluble MIC-A/B: Soluble MIC-A/B not only blocks NKG2D signaling but also attracts immunosuppressive cells, such as myeloid-derived suppressor cells (MDSCs), into the tumor microenvironment, further impairing immune responses [55]. This recruitment is mediated by chemokines and cytokines co-released with soluble MIC-A/B, creating an immunosuppressive niche [56].

NK cells rely also on inhibitory receptors, such as *KIRs* (killer-cell immunoglobulin-like receptors) and *CD94/NKG2A*, to detect self-MHC-I molecules expressed on healthy cells [57]. Tumor cells exploit this system in the following two opposing ways:Upregulation of MHC-I to avoid NK cell activation: Some tumors overexpress self-MHC-I molecules to engage inhibitory receptors on NK cells, reducing their cytotoxic response [58]. This upregulation is mediated by interferon signaling, particularly IFN-γ, which activates the JAK/STAT pathway to enhance MHC-I transcription and presentation on the cell surface [59]. Tumors with mutations in the JAK/STAT pathway can evade this regulatory mechanism, creating a heterogeneous immune evasion strategy, either by over-expressing self-MHC protein or by not expressing at all non-self MHC proteins [60].Downregulation of MHC-I to escape CTLs: To prevent NK cell activation, tumors also upregulate non-classical MHC-I molecules such as HLA-E, which interact with inhibitory receptors like *NKG2A* on NK cells. This adaptation prevents NK cell-mediated cytotoxicity, despite the absence of classical MHC-I [18].

On the other hand, Transforming Growth Factor-Beta (TGF-β) is a potent immunosuppressive cytokine abundantly secreted in the tumor microenvironment (TME) by tumor cells, stromal cells, and regulatory immune cells [61]. Tumor cells increase TGF-β production through hypoxia-induced HIF-1α activity and oncogenic signaling pathways such as SMAD-dependent TGF-β autocrine loops [62]. Mutations in tumor suppressor genes like TP53 can also enhance TGF-β secretion, creating an immunosuppressive microenvironment [63].

There are several mechanisms by which TGF-β contributes to tumor evasion by the immune system. Here, we list some of the most common and important, as follows:Inhibition of NKG2DL expression: TGF-β downregulates NKG2DL on tumor cells’ surface by activating the SMAD2/3 signaling pathway, which represses the transcription of the ligand’s genes. This mechanism is critical for reducing the recognition of NK cells [64].Suppression of effector molecules: TGF-β interferes with the mTOR signaling pathway, reducing the expression of cytotoxic molecules such as granzyme B and perforin. This suppression impairs the ability of NK cells to induce apoptosis in tumor cells [65].Induction of NK cell exhaustion: Chronic exposure to TGF-β leads to an exhausted phenotype in NK cells, characterized by reduced cytokine production (e.g., IFN-γ) and diminished cytotoxicity. This effect is mediated by epigenetic modifications that lock NK cells into a hypofunctional state [66].

We have listed some of the inhibition mechanisms that directly affect NK cells in the TME. However, tumors can also exploit other mechanisms of the immune system that do not directly target NK cells but instead modulate the immune environment. These mechanisms alter the immune microenvironment, weakening NK cell functionality and overall immune responses (Figure 3). Here, we describe some of the indirect contributors to immune evasion, as follows:Production of immunosuppressive molecules: Tumors or associated immune cells in TME (such as TAMs, tumor-associated macrophages) secrete cytokines such as IL-10. The poor oxygen conditions in the TME and the other cells of the tumor mass (f. e. fibroblasts) induce the production of VEGF, which reduces NK cell activity [67]. IL-10 inhibits antigen-presenting cell (APC) maturation, reducing overall immune activation [68]. VEGF not only promotes angiogenesis but also recruits Regulatory T cells (Tregs) and MDSCs, contributing to an immunosuppressive TME [69].Induction of immune checkpoints: Tumor cells upregulate inhibitory ligands such as *PD-L1*, engaging *PD-1* receptors on NK cells and inducing a state of functional anergy [30]. Similarly, the expression of ligands for *LAG-3* and *TIM-3* further suppresses NK cell responses [26].Metabolic constraints in the TME: Hypoxia and nutrient depletion in the TME create metabolic stress on NK cells. High levels of lactate (a byproduct of tumor glycolysis) lower the pH and interfere with NK cell signaling and effector functions. The reduced availability of glucose and amino acids further compromises NK cell metabolism and proliferation [70].

Overall, tumors employ multifaceted strategies to evade NK cell-mediated immunity, from modulating stress ligands to creating immunosuppressive microenvironments. Recent advances, such as the development of cytokine-based therapies and checkpoint inhibitors, pave the way for novel immunotherapeutic strategies that harness the full potential of NK cells. By addressing these challenges, we can improve the efficacy of NK cell-based therapies and provide new hope for patients battling resistant cancers. A combinatory approach integrating cytokine therapies, metabolic reprogramming, and checkpoint inhibitors may offer the most promising avenue for effective cancer immunotherapy targeting NK cells.

#### 2.2.2. Tumor Microenvironment Suppresses NK Cell Activity by Creating an Immunosuppressive Environment

The tumor microenvironment (TME) is a dynamic and active participant in cancer progression [71]. It is composed of cancer cells, endothelial cells, fibroblasts, immune cells, and extracellular matrix components that support tumor growth and enable immune evasion [72]. Evidence shows that NK cells are negatively regulated by several suppressive components of the TME, such as CD4^+^CD25^+^ regulatory T cells (Tregs), antigen-presenting cells (APCs), tumor-associated macrophages (TAMs), and MDSCs. These cell types have been reported to suppress NK cell functions through various mechanisms [73,74].

#### 2.2.3. Downregulation of Activating Receptors

In cancer patients, NK cells often show reduced expression of key activating receptors, such as NKG2D, NKp30, NKp44, Np80, and *DNAM-1* (review in [75]). This reduction in receptor expression significantly impairs NK cell-mediated antitumor immunity. Soluble modulators secreted in TME impaired NK cell antitumor activity, including soluble ligands for activating NK cell receptors, such as *MICA/B* (major histocompatibility complex class I chain-related protein A/B), which bind to NKG2D, leading to its downregulation [76], therefore transforming growth factor beta (TGF-β), extracellular adenosine, prostaglandin E2 (PGE2), and indoleamine 2,3-dioxygenase (IDO).

TGF-β, a cytokine found in the TME, is secreted by various cells, including tumor cells, Tregs, MDSCs, and other stromal cells. This cytokine plays a critical role in suppressing the expansion and activity of effector cells while enhancing the growth of Tregs [61]. TGF-β can impair NK cell function both directly and indirectly, primarily by modulating interactions between NK cells and other cytokine-producing cells [77,78]. For instance, Tregs suppress NK cell proliferation and reduce IFN-γ production by activating TGF-β signaling pathways [79]. Targeting these pathways, such as Tregs in the TME, has emerged as a promising strategy to restore NK cell function and enhance antitumor immunity [80,81]. Ipilimumab, an anti-*CTLA-4*^+^ (cytotoxic T-lymphocyte antigen 4) antibody, has proven to restore NK cell-mediated ADCC in patients with head and neck cancer (HNC). In addition, combining Nivolumab, an anti-programmed death-1 (*PD-1*) antibody, with Mogamulizumab, an anti-C-C chemokine receptor type 4 (*CCR4*) antibody, demonstrated promising antitumor effects [80,81].

Additionally, MDSCs in the TME contribute to the suppression of NK cell activity via membrane-bound TGF-β, which downregulates the expression of NKG2D and IFN-γ in NK cells [82]. NK cell dysfunction is also induced by TAMs through the secretion of TGF-β and the expression of the CTL-4 binding molecules CD80 and CD86 on NK cells [83]. The significant presence of neutrophils in the TME can also suppress IFN-γ production by NK cells by binding NK surface PD-1 receptors to their PDL-1 ligands expressed on neutrophils [84].

Extracellular adenosine, an immunosuppressive metabolite, accumulates in the TME in response to hypoxia via CD39 and CD73. This molecule plays a dual role by suppressing NK cell activity and enhancing the proliferation of Tregs and MDSCs, thereby contributing to the immunosuppressive environment. Adenosine signaling via the A2A adenosine receptor (A2AR) inhibits tumor-infiltrating NK cells [85]. The use of the A2AR antagonist together with the CD73 blockade to reduce adenosine has shown potential to reduce tumor growth and recruit NK cells into the TME [86].

TME also increased the production of prostaglandin E2 (PGE2) as an adopted mechanism to reduce the anti-tumor function of NK cells. NK cell function suppression by PGE2 has been identified in various contexts through the signaling of PGE2 receptor 2 Eprostaonid 2 (Ep2) and receptor 4 (Ep4) [87] by downregulating the expression of NKp30, NKp44, NKp46, and NKG2D, resulting in cytotoxicity inhibition [88]. The cyclooxygenase-2 (COX-2) inhibitor celecoxib has been shown to block PEG2 signaling in many solid tumor clinical trials (https://ClinicalTrials.gov accessed on 2 February 2025) [89]. Another immunosuppressive mechanism in TME involves amino acid metabolism by indoleamine 2,3-dioxigenase (IDO), a rate-limiting enzyme that catalyzed tryptophan to kynurenine, which inhibits NK cell proliferation and reduces NKp46 and NKGD2 on the NK cells’ surface [90,91].

Hypoxia is a hallmark of TMEs and a negative prognostic indicator for numerous solid tumors, causing a reduction in NK cell tumor infiltration but also the resistance of tumor cells to NK cells. As an adaptative mechanism to low oxygen levels, NK cells upregulate HIF-1α and downregulate the expression of activation receptors NKp46, NKp30, NKp44, and NKG2D [92]. HIF-1α also causes NK cells, when stimulated by IL-2 and other cytokines, to lose their ability to upregulate the above activating receptors. Treatment with IL-2 has also been shown to overcome hypoxia-induced NK impairment [93]. Several studies found HIF-1α to be an important NK cell checkpoint [94].

TME is also known for its unique metabolic properties that constitute another key factor to NK cells’ dysfunction. As is widely known, solid tumor cells can generate a variety of metabolic stresses, including poor nutrition and the accumulation of waste products. An example is the lack of glucose and glutamine which directly affects NK cells. Most tumor cells rely on aerobic glycolysis for energy production, which leads to an increase in glucose and glutamine consumption, limiting their use by NK cells and directly affecting their function [95]. While it remains to be seen in the metabolism of NK cells, evidence suggests that glycolysis is increased in activated NK cells compared to oxidative phosphorylation (OXPHOS) in steady-state NK cells [96]. The results of inhibiting both glycolysis and OXPHOS show an impairment in NK cell production of IFN-γ, which translates into attenuated cytotoxicity to leukemia cells [97]. Another effect related to aerobic glycolysis in the tumor cell, also known as the Warburg effect, is the accumulation of lactate in the TME as a metabolic waste product. This results in an acidification of the environment that can impair NK cell function and reduce IFN-γ production by inhibiting the nuclear factor of activated T cells (NFATs) for transcription, a critical transcription factor involved in IFN-γ transcriptional control [98,99]. The TME exerts a profound suppressive influence on NK cell function through a complex interplay of mechanisms, including cytokine signaling, metabolic constraints, and interactions with other immune cells. These findings underscore the importance of understanding these pathways to develop innovative therapies, such as CAR-NK cells and metabolic reprogramming, to overcome the immunosuppressive effects of the TME and enhance cancer immunotherapy.

### 2.3. NK Cell-Based Immunotherapies

Adoptive cell therapy (ACT) is a highly personalized immunotherapy based on the transfer of autologous lymphocytes to patients with advanced malignancies, demonstrating remarkable tumor reduction. While T lymphocytes were the primary focus of ACT for nearly two decades, numerous unsuccessful cases prompted researchers to explore alternative antitumor lymphocytes, leading to the development of NK cell-based immunotherapy. NK cells possess the ability to distinguish between abnormal and healthy cells, providing precise anticancer activity while avoiding adverse complications. This unique characteristic makes them a promising target for ACT [100]. This section reviews various cell sources, expansion and activation methods used to generate NK cells for immunotherapy, the potential of CAR-NK cells, and strategies for combining NK-based therapies.

#### 2.3.1. NK Cell Expansion

Cancer therapies using ready-to-use NK cells have gained significant traction in recent years. Various protocols aim to produce enough functional NK cells, although few meet the high standards of good manufacturing practices (GMPs) essential for clinical application [101,102]. Functional NK cells for immunotherapy can be derived from multiple sources, including umbilical cord blood (UCB), adult peripheral blood (PB), NK cell lines, and bone marrow [101,103,104]. Each source presents distinct advantages and limitations [105]. NK cells derived from UCB are considered an efficient source due to their progenitor population with superior proliferative capacity compared to PB. However, the main limitations include the limited volume of UCB and the low number of NK cells per unit [103]. Both UCB and PB exhibit variability in the NK cell yield caused by donor variability and post-purification efficiency [105]. In contrast, NK cell lines such as the FDA-approved NK92 line provide homogeneous cell populations that expand indefinitely in vitro. NK92 cells exhibit greater cytotoxic activity but require irradiation before administration to patients, which limits their ability to expand and persist in vivo, thereby reducing their anti-tumor activity [101,104,105,106]. Despite their potential in clinical trials, no specific protocols have been defined for their clinical application and their large-scale ex vivo expansion for this purpose remains a challenge [106]. Fu et al. optimized a combination of vitamins in a custom-designed serum-free medium, enhancing the expansion and cytotoxicity of NK cells and providing a solid basis for large-scale ex vivo expansion [106]. As alternative sources of functional NK cells, human embryonic stem cells (hESCs) and induced pluripotent stem cells (iPSCs) represent promising strategies for NK cell-based immunotherapy, as they combine the advantages of other NK cell sources while overcoming their limitations [101,102,104,105]. The indefinite growth capacity of iPSCs and hESCs enables the generation of unlimited, homogeneous NK cells, offering a ready-to-use, standardized approach with enhanced anti-tumor activity. iPSCs also allow for stable genetic modifications that only need to be performed once, after which the modified clone can be expanded to produce a standardized population of NK cells [105].

NK cell expansion requires multiple cellular signals for proliferation, activation, and survival. Protocols to produce large quantities of NK cells rely on feeder cells and culture media optimized with co-stimulatory molecules and cytokine combinations [101,102,103]. The primary feeder cells used include irradiated peripheral blood mononuclear cells (PBMCs), K562 lymphoblastoid cells, and Epstein–Barr virus-transformed lymphoblastoid cells. The K562 cell line provides natural activating signals which, combined with cytokine-generated co-stimulation, enhance NK cell expansion. Genetic modifications of K562 cells have further enabled the expression of cytokines such as IL-21 (mIL-21) and IL-15 (mIL-15) on their membrane. This has led to the generation of several genetically modified K562 lines, including K562-mbIL21, K562-41BBL-mbIL15, K562-OX40L-mbIL18/21, and K562-OX40L [101,103]. However, the use of feeder cells for NK cell expansion presents challenges that require resolution. For instance, prolonged expansion over several weeks may compromise their persistence and cytotoxic function in vivo post-infusion, as the optimal culture conditions provided by feeder cells are no longer present after administration [107].

To address these requirements, Masuyama et al. developed a method for the ex vivo expansion of human NK cells using the co-stimulation of PBMCs with anti-CD3 and anti-CD52 monoclonal antibodies, combined with autologous plasma and IL-2, without the need for feeder cells [107]. It is important to note that the culture medium used for K562 feeder cells contains fetal bovine serum (FBS), human plasma, or human AB serum, raising concerns about infection risks. Additionally, NK cell expansion systems using K562 cells remain unoptimized. To enhance safety in clinical applications of NK cell expansion with tumor cell lines, feeder cell-based systems employing irradiated cells, such as irradiated PBMCs, have been developed. These systems have proven advantageous, achieving an average NK cell expansion of 15,000-fold on day 14 with a purity of 98.2 ± 0.3% [104,108].

Although there is no consensus on the optimal medium for NK cell expansion, it is evident that the culture medium and the supplements used significantly influence the expansion and functionality of these cells [103]. Currently, NK cells are predominantly expanded in Roswell Park Memorial Institute (RPMI) 1640 medium supplemented with FBS (RPMI complete; RC) [103,109]. Another study demonstrated that Dulbecco’s Modified Eagle Medium (DMEM), enriched with Ham’s F12 Nutrient Mix and supplemented with human AB serum, β-mercaptoethanol, GlutaMAX, ascorbic acid, and ethanolamine, further enhanced NK cell proliferation compared to RC medium [103,109,110]. Koh et al. [103] compared three basal media: RPMI-1640 and Dulbecco’s Modified Eagle Medium (DMEM) (Thermo Fisher Scientific), both supplemented with serum and serum-free alternatives. They found that while serum supplementation enhanced the rate of NK cell expansion, it reduced cytotoxicity. In contrast, serum-free medium increased cytotoxic capacity. Based on these observations, they developed a new culture protocol that begins with serum-supplemented DMEM and transitions to serum-free medium on day 14. This optimized approach shows promise as an alternative to serum-containing media for expanding NK cells in immunotherapy applications [103,109]. Regarding soluble factors, co-stimulation with cytokine cocktails has been extensively studied to enhance NK cell expansion, enabling them to acquire a broader range of effector functions [105,111]. The most used cytokines include interleukin-2 (IL-2), IL-15, IL-18, IL-21, and IL-27. Additionally, the combination of IL-12, IL-15, and IL-18 generates a phenotype known as cytokine-induced memory (CIML) NK cells, which exhibit enhanced survival, expansion, and anergy avoidance in vivo when incubated with donor-derived NK cells [105,112,113,114]. CIML NK cells further enhance NK cell therapies by incorporating CAR gene constructs and/or combining them with immunomodulatory agents [114].

The therapeutic efficacy of expanded NK cells also depends on their viability and functionality after thawing. Thawed NK cells undergo significant changes in cytokine production, cytotoxic activity, proliferation, and in vivo migration [115]. Consequently, the adaptation of protocols to clinical-grade production must consider not only good manufacturing practices (GMPs) but also optimized freezing conditions to establish viable NK cell stocks. Berjis et al. [116] demonstrated that pre-treating NK cells with a combination of IL-15 and IL-18 before cryopreservation significantly improved their recovery rates to ~90–100% [102,116].

One of the main challenges in NK cell-based immunotherapy is producing large quantities of NK cells for clinical applications. Developing large-scale expansion methods is crucial to advancing their use in both clinical settings and translational research laboratories [104,117]. Large-scale production requires closed and automated systems that ensure GMP compliance while also offering significant long-term cost savings. Typically, large-scale NK cell production takes 14–21 days. In contrast, traditional manual production demands constant replenishment of fresh medium and cytokines, which increases the risk of contamination [104,117]. Recently, large-scale closed NK cell expansion systems, such as the magnetically controlled bioreactor with a floating magnetic mixer, have been developed. This bioreactor, controlled by an intermittent magnetic field, successfully cultured NK-92 cells, achieving a more efficient expansion than traditional culture flasks [118]. Wang et al. developed G-Rex 100 M bioreactors based on gas-permeable membrane technology. This platform facilitates the production of ready-to-use NK cells, supporting their future clinical implementation in immunotherapy [117]. Despite all these advances, the development of automated and closed NK cell expansion systems remains a challenge and a large field of study due to the wide variety of NK cell sources and the different expansion protocols used.

#### 2.3.2. CAR-NK Cells

CAR-T cell therapy has revolutionized the treatment of hematologic malignancies over the last decade, achieving significant success with the approval of therapies targeting diseases such as B-cell acute lymphoblastic leukemia, large B-cell lymphoma, follicular lymphoma, and multiple myeloma. Despite its success, challenges remain in its limited efficacy against solid tumors [119,120,121,122]. Next-generation CAR-based therapies such as CAR-NK cells indicate solutions for patients with advanced or metastatic solid tumors. For instance, CAR-NK cells have shown significant potential in preclinical studies targeting ovarian and gastric cancers, where traditional CAR-T cell therapies have been less effective due to the immunosuppressive tumor microenvironment. However, optimizing CAR-NK therapies for effective use in solid tumor immunotherapy remains a key focus of ongoing research [123].

Chimeric antigen receptors (CARs) are artificially engineered fusion proteins designed to enhance immune cell targeting of tumors. Much like CAR-T cells, the functional CAR molecule expressed on NK cells is composed of three elements: an extracellular domain, a transmembrane region, and an intracellular signaling domain. First-generation CARs featured a simple structure, consisting of a single-chain variable fragment (scFv) for antigen recognition and a CD3 ζ-chain for intracellular signaling. While first-generation CARs provided a basic framework for tumor targeting, subsequent generations introduced co-stimulatory domains such as 4-1BB and CD28. These advancements enhanced cell activation, improved proliferation, and increased survival rates, making CAR-NK cells more effective in tumor elimination. Later generations integrated co-stimulatory molecules like 4-1BB, CD28, and OX40, enhancing cell activation, proliferation, and survival rates. Fourth-generation CARs, known as armored or precision CARs, release immune modulators in the tumor microenvironment, while fifth-generation CARs incorporate advanced binding motifs for precision therapy [123,124,125]. To date, the second-generation CAR structures CD28-CD3ζ and 41BB-CD3ζ and the third generation CD28-41BB-CD3ζ are the most used in CAR-NK cells [125].

The NK cells used in NK cell engineering can be derived from several primary sources: peripheral blood (PB), induced pluripotent stem cells (iPSCs), unstimulated leukapheresis products (PBSCs), bone marrow (BM), human embryonic stem cells (hESCs), umbilical cord blood (UCB), mesenchymal stem cells, or established NK cell lines such as NK-92. These cells, a subset of lymphocytes, exhibit the unique ability to recognize and eliminate tumor cells without prior sensitization, antibody involvement, or MHC restriction. Common markers for NK cells include CD16 and CD56. The characteristics, advantages, and limitations of NK cells derived from these sources play a crucial role in their clinical application and therapeutic potential [124,126].

CAR-NK cells bring numerous advantages to cell-based immunotherapy, including their compatibility with donor-derived sources for manufacturing, multiple mechanisms of cytotoxicity, and a significantly reduced risk of toxicities. The risk of on-target/off-tumor toxicity to healthy tissues is relatively low because CAR NK cells have a short lifespan in circulation [127]. CAR NK therapy eliminates the need for using autologous NK cells, as these cells present a lower risk of alloreactivity and graft-versus-host disease (GvHD). Allogeneic NK cells are considered safe for adoptive cell therapy (ACT) and overcome one of the principal limitations that autologous T cells in CAR-T cell therapy present in their production. NK cell sources are readily available in clinical samples and, as mentioned above, can be derived from a variety of origins and are therefore a viable large-scale production of “off-the-shelf” CAR-NK cells [128,129].

One major advantage of CAR-NK cell therapy is its superior safety profile, with a markedly lower incidence of cytokine release syndrome (CRS) and neurotoxicity compared to CAR-T therapy [130,131]. This difference is likely due to the unique cytokine secretion profile of CAR-NK cells, as NK cells produce only minimal amounts of IFN-γ and granulocyte-macrophage colony-stimulating factor (GM-CSF) with reduced levels of IL-1β, IL-2, and IL-6, key triggers of CRS [132,133]. NK cells not only target cancer cells via single-chain antibody recognition of tumor surface antigens but also detect various ligands through multiple receptors, including *NCRs, NKG2D*, co-stimulatory receptor *DNAM-1* (*CD226*), and some activating *KIRs* [134]. Compared to CAR-T cells, CAR-NK cells exhibit superior tumor-specific targeting and cytotoxic potential. Their ability to exploit both CAR- and NK receptor-dependent mechanisms positions them as a highly effective strategy for eradicating heterogeneous tumors, even in cases where target antigens are absent. Even in tumor cells lacking the CAR target antigen, both CAR- and NK cell receptor-dependent mechanisms can be exploited [128]. Looking ahead, CAR-NK cells are poised to become a cornerstone in cancer immunotherapy, combining precision targeting with enhanced safety and scalability. Their versatility in targeting both hematologic and solid malignancies highlight their transformative potential in reshaping the treatment landscape.

#### 2.3.3. Combined NK-Based Immunotherapies

The rise of NK cell-based therapies has spurred the development of new combination approaches designed to enhance the functional characteristics of NK cells, enabling stronger and more durable anti-tumor effects. Notably, combination therapies using non-genetically modified NK cells are more prevalent in hematological malignancies (62%) than in solid tumors (38%), whereas genetically modified NK cells are predominantly applied to solid tumors (76%) compared to hematological malignancies (24%) [135].

One strategy to achieve synergistic effects and enhance the efficacy of NK cell therapies involves the integration of small molecules and antibodies that counteract metabolite-mediated immunosuppression in the tumor microenvironment (TME) or target immune checkpoints [136]. Immune checkpoint inhibitors (ICIs), such as anti-*PD-L1*, *PD-1*, and *CTLA-4* antibodies, have demonstrated the ability to stimulate the immune system to effectively eliminate tumors [13]. Pembrolizumab and Nivolumab, anti-PD-1 monoclonal antibodies (mAbs), are used to target solid tumors in combination with unmodified NK cells (PB-NKs), iPSC-derived NK cells, and engineered iPSC-derived NK products, such as FT538 and ImmunityBio’s high-affinity NK (t-haNK) cells [135]. Fabian KP, Padget MR, Donahue RN, et al. developed, for the first time, a PD-L1-targeted t-haNK cell line derived from NK-92 cells, engineered to express the high-affinity CD16 allele, endoplasmic reticulum-retained IL-2, and a PD-L1-specific CAR. The anti-tumor activity of t-haNK PD-L1 cells was evaluated in combination with anti-PD-1 and an IL-15 superagonist (N-803). The results of this study support the potential use of these cells in clinical trials [137]. Ipilimumab, an anti-CTLA-4 monoclonal antibody (mAb), is used in patients with head and neck squamous cell carcinoma in combination with non-genetically modified allogeneic CIML NK cells [135]. Numerous therapies have been preclinically tested in combination with CAR-NK cells. A phase II clinical trial is underway to evaluate the effectiveness of a PD-L1-targeted CAR-NK combination therapy with Pembrolizumab and N-803 in patients with recurrent or metastatic gastric or head and neck cancer (NCT04847466). Another promising candidate for combination with CAR-NK cells is lenalidomide, which induces the expression of ligands on cancer cells recognized by NK cells. It has been tested in studies involving neuroblastoma and hematopoietic malignancies with expanded and activated unmodified NK cells (NCT02573896, NCT02481934, and NCT02280525) [138]. Strecker et al. evaluated a local combination therapy targeting glioblastoma. This approach utilized an anti-PD-1 immunoadhesin delivered via an adeno-associated viral vector (AAV) combined with specific HER2 CAR-NK92 cells. The results demonstrated that this represents a novel and promising strategy for glioma immunotherapy. Thanks to the high flexibility in the selection of the target CAR-NK cells and AVV, this immunotherapy has the potential to be customized against particular malignant cells and a specific TME [139]. Other inhibitory receptors expressed in tumor-infiltrating NK cells include LAG-3, TIM-3, and TIGIT. Efforts are underway to explore how targeting these immune checkpoints in combination with CAR-NK cells could enhance NK-mediated anti-tumor activity [13,140].

Additional combination strategies involve molecular inhibitors, encompassing a variety of therapeutic agents with distinct mechanisms of action, such as tyrosine kinase inhibitors that disrupt signal transduction pathways of protein kinases [135]. For instance, Regorafenib combined with CAR-NK92 cells targeting EpCAM demonstrated superior anti-tumor activity compared to monotherapy in colorectal cancer treatments [138,141]. Similarly, a synergistic effect was observed with the combination of Cabozantinib and CAR-NK92 cells targeting EGFR in renal cell carcinoma in vitro [138,142].

Recent studies have demonstrated that radiotherapy (RT) not only directly kills tumor cells but also enhances the tumor immune microenvironment, boosting the immune system’s anti-tumor response by altering molecular expression in tumor cells and releasing tumor antigens [100,143]. The ‘abscopal effect’ of RT may synergize with immunotherapy, making the combination of ACT and RT a promising approach for enhancing tumor responses [143]. Additionally, NK cells play a complementary role with RT, as the infusion of mature NK cells can mitigate the persistent radioresistance of solid tumors, rendering them more radiosensitive [100]. Lin et al. demonstrated in a preclinical model of hepatocellular carcinoma a significant enhancement of the anti-tumor effect of CAR-NK cells following a single high dose of irradiation [143]. Another study demonstrated the potential clinical benefit of incorporating the PD-L1/PD-1 checkpoint blockade into radiotherapy regimens to sensitize tumor cells to NK cell-mediated killing in nasopharyngeal carcinoma [144]. Low-dose chemotherapy exhibits an immunomodulatory role by triggering tumor antigen presentation and activating dendritic cells [121]. Several studies have reported that, in the presence of cisplatin, CAR-T/NK cells retain their cytotoxicity, which is further enhanced under these conditions [140]. Notably, the sequential combination of cisplatin with CD44-CAR-NK92 and CD133-CAR-NK cells demonstrated the strongest anti-tumor effect on ovarian cancer stem cell lines [121,140,145,146].

To reduce the toxic effects of chemotherapy, hepatic arterial infusion chemotherapy (HAIC) is applied in the treatment of locally advanced hepatocellular carcinoma (HCC). A phase I study shows that the combination of NK cells with 5-fluorouracil (5-FU) and cisplatin HAIC is a promising therapy for patients with locally advanced HCC [147]. Additionally, the potential of combining chemoradiotherapy (CRT) with NK cell-mediated immune responses has been investigated [148]. For example, NK cells targeting membrane heat shock protein 70 were combined with CRT and the subsequent PD-1 checkpoint blockade in non-small cell lung carcinoma (NSCLC), enhancing immune cell efficacy [149]. A phase II randomized controlled trial reported favorable results using autologous NK cells following chemoradiotherapy in NSCLC [150,151]. Nguyen et al. conducted the first study combining low-dose radiotherapy, 5-FU, and NK cell immunotherapy in intractable colorectal cancer. The results suggest that this combination therapy maximizes anti-tumor effects against colorectal cancer [148].

Local ablative therapies, such as photothermal therapy or microwave ablation, induce the release of immunomodulatory factors, including tumor antigens and cytokines that stimulate antitumoral immune responses. Additionally, they generate hyperthermic damage to the tumor, reducing its dense structure and interstitial fluid pressure [121,140]. Combining these therapies with ACT has the potential to improve the therapeutic index of NK cells in solid tumors [140]. For instance, CAR-NK cell immunotherapy combined with photothermal therapy, using an NIR-II pentamodal-guided temperature feedback nanoplatform, demonstrated a promising therapeutic effect in lung cancer [140,152].

Thus, various combination therapy approaches can synergistically enhance the efficacy of NK cell-based ACTs (Figure 4). Furthermore, advances in cell engineering techniques have the potential to address key limitations in solid tumors, including the restricted persistence and effector function of NK cells within an immunosuppressive TME, as well as poor tumor trafficking.

### 2.4. Clinical Trials and Recent Results

Over the last decade, immunotherapy has brought significant advancements to clinical oncology. Notably, immune checkpoint inhibitors and CAR-T cells have demonstrated remarkable efficacy across various malignancies. Nevertheless, these treatments benefit only a fraction of patients, leaving substantial medical needs unmet [153,154].

NK cell therapies are revolutionizing cancer treatment, as recent clinical trials reveal their promising outcomes and potential in immunotherapy. Over the past few years, there has been a growing interest in these therapies, reflected in the increasing number of global clinical trials since the first CAR-NK-cell clinical trials (NCT00995137, clinicaltrials.gov) started in 2009. Although the field is still in its early stages, with most trials being Phase I and Phase I/II studies, several Phase II trials and a few advanced studies (Phase II/III, III, or IV) have already been conducted [135,155] (see Table 1). The clinical outcomes observed to date make a strong case for the ongoing exploration and advancement of NK cells as a key element in the future of immunotherapeutic strategies [156]. As a result, it is expected that in the coming years, more therapies will progress to pivotal clinical stages, with market authorization likely on the horizon [157,158].

Clinical trials using NK cell-based therapies for hematological cancers, such as Acute Myeloid Leukemia (AML), have demonstrated both safety and efficacy for treatments involving allogeneic or autologous NK cells derived from donors [159]. As efforts continue to enhance treatment outcomes and develop more scalable methods for producing NK cells for clinical use, new trials are emerging to assess these advancements and broaden their potential applications. NK cells have been modified to express CARs, enabling them to target B-cell malignancies effectively. Like CAR-T cell therapies, the majority of CAR-NK cell studies focus on markers associated with hematopoietic malignancies, including CD19, CD20, CD22, and BCMA. Among these, CD19 remains the primary target in both preclinical and clinical research efforts to date [160].

A pioneering trial with CAR NK-92 cells, involving three patients with relapsed or refractory AML, revealed that these cells could be infused at doses up to 5 billion per patient with minimal adverse effects [161]. Additionally, a significant phase I/II trial (NCT03056339) carried out at MD Anderson Cancer Center is underway using cord blood-derived CAR-NK cells targeting CD19 in patients with relapsed or refractory CD19^+^ cancers, such as non-Hodgkin’s lymphoma and chronic lymphocytic leukemia. The preliminary results show that eight out of eleven patients responded positively to the treatment, with no significant toxic effects observed [162]. Thus, the administration of CAR-NK cells did not result in the onset of cytokine release syndrome, neurotoxicity, or graft-versus-host disease, nor was there any rise in inflammatory cytokine levels, including interleukin-6, above baseline levels. A maximum tolerated dose was not reached, even with doses as high as 1 × 10^7^ CD19-targeted CAR-NK cells per kilogram of body weight. Furthermore, CAR-NK cells were found to remain in the peripheral blood of patients for a minimum of 12 months post-infusion. While these results were not influenced by dose or HLA mismatch, early cell expansion was observed to correlate with positive treatment outcomes. [138]. This encouraging response underscores the potential for expanding the use of this therapy in a broader clinical context [162,163].

An increasing number of in vitro and in vivo studies have examined the activity of CAR-NK cells against solid tumors, with glioblastoma, breast cancer, and ovarian cancer being the most widely researched to determine their therapeutic potential. While these cells show unique advantages, the shift of CAR-NK cell therapy from blood cancers to solid tumors has faced technical and biological challenges [164]. These obstacles include issues related to cell persistence, overcoming the immunosuppressive microenvironment, and optimizing transduction efficiency, among others [165]. Advancements, such as bifunctional lipid nanoparticles (DLNPs) and peptide-based CAR-NK cells, are paving the way to overcome challenges posed by the immunosuppressive tumor microenvironment. These innovations represent a significant step toward improving the effectiveness of CAR-NK cell therapies against solid tumors.

For instance, DLNPs have been designed to enhance the antitumor efficacy of CAR-NK cells by facilitating their activation and efficient CAR mRNA delivery. Recent studies have highlighted the potential of DLNPs to improve CAR-NK cell therapy in solid tumors, demonstrating promising results in preclinical settings [123,158,160,166]. On the other hand, Wang et al. recently introduced peptide-based CAR-NK cells as a novel strategy for the treatment of solid tumors. This approach offers a complementary avenue for addressing the challenges of tumor heterogeneity and immunosuppression within the tumor microenvironment [167].

Building on these promising results, current efforts have expanded to include trials targeting a variety of solid tumors. To date, there are 25 clinical trials of CAR-NK applied to solid tumors [123]. Although only a limited number of clinical trials have evaluated CAR-NK products based on NK92, PB-NK, and UCB-NK cells, there is growing interest in targeting common tumor antigens like Roundabout homolog 1 (ROBO1), NK cell-activating receptor (*NKG2D*), MSLN, HER2, and MUC1. Several clinical trials have tested human primary NK cells modified with CARs targeting specific tumor antigens, including ROBO1 for solid tumors (NCT03940820), PSMA for prostate cancer (NCT03692663), MSLN for epithelial ovarian cancer (NCT03692637), and Claudin6 for ovarian, testicular, and refractory endometrial cancers (NCT05410717). Additionally, trials investigating CAR-NK92 therapy are underway, including studies targeting HER2 for glioblastoma (NCT03383978) and chimeric costimulatory converting receptor (CCCR) CAR-NK for non-small cell lung cancer (NCT03656705). MUC1-specific CAR-NK cells are being developed to treat relapsed or refractory solid tumors (NCT02839954), with promising results showing stable disease in seven of eight evaluable patients without serious adverse events. A combination therapy using anti-ROBO1-specific biCAR-NK-92 for pancreatic cancer has also been evaluated (NCT03941457), while another phase-I trial (NCT05528341) is exploring the use of NKG2D CAR-NK92 cells for relapsed or refractory solid tumors. Another phase-I clinical trial (NCT03415100) is recruiting patients with metastatic solid tumors. The study focuses on assessing the safety of NKG2DL-targeting CAR-NK cells, transfected by mRNA electroporation. Two additional early-phase-I trials (NCT05137275 and NCT05194709) are focusing on targeting the 5T4 oncofetal antigen in locally advanced or metastatic solid tumors using CAR-NK cells. The FDA has recently approved an investigational application for FT536, a CAR-NK cell therapy developed by Fate Therapeutics (NCT05395052), aimed at treating patients with advanced solid tumors. FT536 is an allogeneic, engineered-induced pluripotent stem cell-derived NK cell therapy, genetically modified to target the alpha-3 domain of the MHC class I-related proteins-A (MICA) and -B (MICB) [121,168].

To fully integrate CAR-NK therapy into future clinical treatments, it is essential to validate the foundational experimental findings of CAR-NK in various solid tumors, as discussed in earlier clinical trials. Tailored recommendations for addressing distinct tumor microenvironments (TMEs) within the human body must also be made. Alongside this, specific optimizations should be carried out, while continuously monitoring and documenting any side effects that arise during treatment.

It is known that the application of CAR-NK therapy for solid tumors is heavily impacted by the TME. Various factors such as hypoxia, an acidic milieu, elevated adenosine levels, NK cell surface receptors, TGF-β, interleukins, and exosomes all contribute to modulating the anti-tumor effects of CAR-NK cells. These therapies can hinder immune evasion by cancer cells by targeting intrinsic pathways, counteracting tumor heterogeneity, neutralizing the inhibitory elements of the TME, and enhancing CAR-NK cell infiltration into tumor tissues. Despite ongoing clinical trials, the understanding of the mechanisms behind CAR-NK’s effectiveness in solid tumors, especially in liver cancer, cholangiocarcinoma, and urological cancers, remains limited. Many of these trials are still in their early stages, with small participant groups and considerable limitations. As these clinical trials progress, it is expected that accumulating data will further demonstrate the clinical potential of CAR-NK in treating solid tumors.

In short, ongoing clinical studies investigating the safety and efficacy of CAR-NK cell therapy in both hematological and solid cancers highlight its potential to reshape cancer treatment strategies. Additional research and clinical investigations are essential to fully comprehend and utilize the transformative capabilities of CAR-NK cells in immunotherapy.

As clinical trials advance, CAR-NK therapies are poised to redefine cancer treatment, offering groundbreaking solutions for both hematological and solid tumors.

The following Table 2 provides an overview of ongoing clinical trials involving CAR-NK cells, summarizing their targets, cell sources, and clinical applications across different types of malignancies.

These trials exemplify the growing interest and commitment to developing CAR-NK therapies as a cornerstone of future cancer treatments. This summary highlights the ongoing efforts to harness the full potential of CAR-NK therapies, underscoring their transformative role in cancer immunotherapy.

In addition to the CAR strategy, other cancer immunotherapy approaches include oncolytic viruses and immune checkpoint inhibitors. These therapies—CAR-NK cells, oncolytic viruses, and immune checkpoint inhibitors—represent some of the most advanced treatments available, each with distinct mechanisms and uses [169,170]. CAR-NK cells, engineered to target specific cancer-related antigens, are particularly valued for their safety and precision, especially in hematologic cancers. Oncolytic viruses, which selectively infect and destroy cancer cells, show potential in treating solid tumors, although their effectiveness can vary [171]. Immune checkpoint inhibitors work by boosting the immune system’s ability to combat cancer and have proven effective in a range of cancers, although they can lead to significant immune-related side effects [172]. While CAR-NK cells are still in the early stages of development and their production is complex, oncolytic viruses require careful genetic modification to ensure effectiveness. Immune checkpoint inhibitors, although widely applicable, face challenges related to cost and accessibility. Each of these therapies has its own advantages and challenges, influencing treatment plans that are tailored to the cancer type, stage, and the patient’s overall health. As research progresses, the landscape of these treatments continues to expand, offering new possibilities in cancer therapy.

### 2.5. Challenges and Future Directions

#### 2.5.1. Principal Challenges of NK Cell Therapies

##### Short Lifespan

The limited lifespan of NK cells represents a major obstacle to their clinical efficacy, as rapid apoptosis is often induced by signals within the tumor microenvironment. Pro-inflammatory cytokines and other stressors activate apoptotic pathways, further compromising NK cell survival [163,173,174]. Current strategies aim to counteract these effects by modulating the tumor microenvironment and enhancing NK cell resistance to apoptosis [163,175].

Therefore, the tumor microenvironment plays a crucial and determining role in regulating both the lifespan and functionality of NK cells [163,174]. This cellular environment is characterized by the presence of various immunosuppressive cells and factors that negatively impact NK cell activity [163,174]. Tumor cells, in their efforts to evade the immune response, can secrete interleukins such as IL-10 and TGF-β, which act as potent inhibitors of NK cell proliferation and effectiveness [174,176]. Furthermore, it has been demonstrated that hypoxia present in the tumor microenvironment significantly affects the functionality of NK cells, thereby contributing to their short lifespan and response capacity [163,176]. Collectively, these factors hinder the efforts of NK cells, creating an environment that favors the growth and survival of tumor cells at the expense of normal immune efficacy.

##### Expansion and Activation

Expanding and activating NK cells effectively is another significant challenge in clinical applications [175,177]. Cultivating sufficient viable and functional NK cells requires complex protocols involving cytokines like IL-15 and genetically modified feeder cells, both of which have shown promise in enhancing NK cell proliferation and cytotoxicity [163,175,177]. However, these methods require further optimization to meet clinical safety and scalability standards [177,178]. NK cells’ limited capacity for ex vivo expansion hinders the production of adequate quantities for effective treatments [163,175,177,178].

Various strategies have been rigorously explored and systematically analyzed to enhance the expansion of NK cells, featuring the use of cytokines such as IL-15, a factor that has been shown to significantly increase both the proliferation and cytotoxic activity of NK cells in several recent studies [174,176,177,179]. Furthermore, the incorporation of genetically modified feeder cells has proven to be highly effective and promising in enhancing the expansion of NK cells, suggesting a truly encouraging pathway for the development of improved cell therapies in the future [124,175,176,177,179].

Nevertheless, despite advancements, these innovative techniques require a considerable degree of standardization and ongoing optimization to ensure their effective clinical application and safety within the current therapeutic landscape and for future disease treatments [177,178,180].

While challenges in NK cell expansion remain, the integration of advanced engineering techniques, such as CAR constructs, is addressing these limitations and opening new therapeutic avenues.

To better understand the differences, advantages, and limitations of NK cell therapies compared to CAR-T and CAR-NK therapies, the following Table 3 summarizes their key characteristics based on recent scientific insights.

##### Mechanisms of Tumor Resistance

These mechanisms represent a critical and challenging obstacle in NK cell therapy, limiting the effectiveness of these treatments. It is known that tumor cells have developed the ability to evade immune surveillance through various sophisticated strategies. Tumor cells evade NK cells by downregulating activation ligands critical for immune recognition [163,174,176]. Additionally, tumor cells can also secrete immunosuppressive factors that significantly restrict the ability of NK cells to recognize, attack, and effectively destroy cancer cells. This leads to a series of complications within management of therapy, making the fight against cancer even more arduous and complex. This is one of the reasons why many recent studies have addressed these aspects [163,175,181], highlighting the importance of understanding these details to advance better therapeutic approaches.

Regulatory T cells and tumor-associated macrophages dominate the tumor microenvironment, suppressing NK cell activity [155,178]. These complex and diverse interactions can induce a state of functional exhaustion in NK cells, which in turn clearly limits their ability to respond effectively to the presence of tumor cells [155,163,178]. This implies that the tumor microenvironment does not merely act as a passive setting; rather, it plays an active role in modulating and restricting the immune response, resulting in difficulties within the identification and elimination of malignant cells by the immune system [163,174,176].

##### Areas of Future Research

Emerging areas of research hold great promise for improving NK cell therapies. Gene-editing technologies such as CRISPR are enabling precise modifications to enhance NK cell persistence and functionality, overcoming tumor resistance mechanisms. Additionally, CAR-NK cells are being developed to target specific tumor antigens, offering a personalized and highly effective immunotherapy option [182,183,184].

#### 2.5.2. Overcoming the Limitations: Strategies

To address the major limitations associated with NK cell-based therapies, several strategies have been developed to improve their expansion, persistence, cytotoxicity, and tumor infiltration.

##### Gene Editing by CRISPR

One of the most promising areas is undoubtedly the use of CRISPR gene editing to modify NK cells with the aim of enabling these cells to overcome the various resistance mechanisms that commonly arise in treatments. This is crucial for improving the effectiveness of therapies [182,183,184]. This advanced technology allows for the specific and precise modification of genes that regulate both the activation and persistence of NK cells within the body, which could substantially increase their efficacy in cancer treatment [182,183,184,185].

Moreover, combining these cells with immunomodulators, featuring immune checkpoint inhibitors and cytokine therapies, has the potential to further enhance their effectiveness. Consequently, NK cells are currently being evaluated in clinical trials, with those derived from umbilical cord blood and stem cells demonstrating considerable promises for adoptive cell therapies [163,178,183].

As another example, the impact of deleting the CISH gene in human primary natural killer (NK) cells using CRISPR/Cas9 technology has been reported [186]. Interestingly, this deletion of CISH, which encodes the protein CIS—a known negative regulator of NK cell activity—aimed to enhance the anti-tumor efficacy of NK cells against glioblastoma (GBM), a highly aggressive brain tumor [186].

Another study generated CD38 gene knockout (KO) NK cells, which demonstrated resistance to antibody-induced fratricide and improved their ability to mediate antitumor responses [187]. Furthermore, the introduction of a high-affinity CD16 receptor into the CD38 locus significantly boosted the antitumor activity of NK cells against multiple myeloma in both in vitro and in vivo models [187]. Therefore, these findings highlight the potential of CRISPR/Cas9-mediated gene editing to improve the efficacy and scalability of NK cell-based therapies, offering a promising approach to advancing immunotherapy for hematological malignancies [185,186,187].

Thanks to genetic engineering, it has been shown that NK cells can be useful in the treatment of infectious diseases, as they play a crucial role in innate immunity [163,178,183,185]. This is due to their essential role in the elimination of infected cells and in the production of cytokines that target viruses such as HIV-1 and SARS-CoV-2, among others [183,185,188,189].

In a nutshell, all these studies provide evidence that continuing to explore these possibilities has the potential to significantly improve patient outcomes, offering new hope in the field of oncology.

##### CAR-NK Therapies

Conversely, the development of innovative CAR-NK therapies has demonstrated remarkable potential in treating various complex diseases. These immune cells can be engineered to specifically target antigens present on malignant and aggressive tumor cells [124,138,155,176,178,181], allowing more tailored treatments and, as a result, improving therapeutic efficacy.

Future research will focus on identifying specific biomarkers to guide treatment selection and developing advanced technologies for the delivery of targeted therapies. However, optimizing the structure of CAR and the culture conditions is crucial to maximizing the antitumor activity and persistence of NK cells within the body [124,138,155,176,178,181,190].

Recent studies have also shown that CAR-NK cells can be efficiently produced using induced pluripotent stem cells (iPSCs), offering an innovative, sustainable, and scalable approach to cell therapy [155,181,190,191]. This method holds promise for major advancements in regenerative medicine and opens new avenues for treating several diseases [155,181,190,191].

Furthermore, recent articles highlight the potential of mRNA-modified CAR-NK cells as a safe and effective therapy for pediatric sarcomas. These cells, targeting the EphA2 receptor, demonstrated high specificity and efficacy in preclinical models, thanks to advancements such as improved mRNA stability and transient expression, which minimize genetic risks [192]. Although challenges remain, such as optimizing their persistence in solid tumors, CAR-NK cells are emerging as a scalable and promising solution in immunotherapy, with the potential to revolutionize the treatment of difficult-to-cure diseases [138,182,192].

A key advantage of CAR-NK cells is their fast production and expansion capacity, enabling quicker preparation compared to modified T cells [124,138,155,176,178,181]. Additionally, NK cells have a favorable safety profile, with fewer side effects related to cytokine release syndrome compared to CAR-T therapies [124,138,155,174,176,178,181].

In addition, high-performance cell expansion protocols, including perfusion bioreactors, are revolutionizing the efficient large-scale production of NK cells with enhanced cytotoxic activity [193]. These advanced systems allow precise control over culture conditions, improving cell yield and functionality while reducing production time and costs. As these technologies continue to evolve and manufacturing processes are further optimized, the future of NK cells in biomedicine appears increasingly promising, paving the way for more personalized and effective cancer treatments [124,138,155,163,175,177,179,180,182,184].

Despite these advancements, challenges remain and researching genetic editing strategies, such as CRISPR, are critical areas of focus [124,163,175,176,178,181] in order to harness the full potential of CAR-NK-based therapies and further developments in this promising field of investigation [182,184,194].

##### Most Recent Studies

Recent studies report that CAR-NK cells, particularly those derived from umbilical cord blood, offer promising potential as universal, ready-to-use treatments, eliminating the need for customized T cells [155,191,195]. Some studies highlight the potential for improving their design to extend their persistence and enhance their efficacy using advanced cytokines, such as soluble IL-15 [174,176,179,195].

Similarly, recent reports highlight how peptide-based CAR-NK cells offer key advantages, such as the mitigation of off-target toxicity and enhanced efficacy in immunosuppressive tumor microenvironments [124,138,155,167,176,178,181,191]. Published designs of specific CAR-NK cells use multiple peptides as targeting molecules, allowing for the simultaneous targeting of tumor-associated antigens (TAAs) featuring PD-L1, EGFR, and VEGFR2 [167]. This approach increases their tumor-killing capacity while reducing unwanted side effects and toxicity, demonstrating their effectiveness in both in vitro and in vivo studies [167,191].

Another example of CAR-modified NK cells has been reported lately by targeting ErbB3, a key receptor in breast cancer progression and resistance [196]. These CAR-NK cells demonstrated high cytotoxicity against ErbB3-positive tumor cells and significantly reduced tumor size in animal models, without observable side effects [196]. Although their infiltration into solid tumors was limited, recent results underscored their promise as a therapy for resistant cancers, with ongoing challenges in clinical optimization [182,196].

These studies provide evidence of how research is continuously advancing with an increasing number of studies discussing the promising future of NK cells in immunotherapeutic treatments, such as in the management of head and neck squamous cell carcinoma (HNSCC) [28]. Curiously, it has recently been reported that the TIM-3 receptor is a key target, as its blockade can enhance the cytotoxic functions of NK cells [28,197]. Furthermore, the galectin-9 ligand plays a crucial role in suppressing NK cell cytotoxicity and proliferation, suggesting that strategies specifically targeting this ligand could revitalize the antitumor potential of NK cells [197].

Despite the challenges, advancements in genetic engineering, CAR-NK design, and innovative expansion protocols are driving the evolution of NK cell-based therapies. These breakthroughs hold the potential to transform oncology by providing safer, more efficient, and accessible treatment options for patients worldwide. On the one hand, advances in CRISPR gene editing are opening new possibilities to enhance the functionality and persistence of NK cells, allowing them to overcome tumor resistance mechanisms. On the other hand, the development of CAR-NK therapies has shown significant potential, with ongoing efforts to improve their design and optimize production processes. Therefore, further investigation of these areas holds great promise for the future of cancer immunotherapy, offering hope for more effective and personalized treatments in the fight against cancer.

These advancements represent a milestone in immunotherapy, providing hope for patients and establishing a new standard in cancer treatment.

## 3. Summary and Future Perspective

Recent advances in NK cell-based therapies have significantly enhanced their potential in cancer immunotherapy. While adoptive NK cell transfer and CAR-NK cell engineering have shown promising results, several challenges persist, particularly in the context of persistence and immunosuppressive tumor microenvironments.

Studies highlight that tumor cells evade NK cell-mediated immunity by downregulating activating ligands, overexpressing inhibitory molecules, and altering metabolic conditions to suppress NK function [4,5,7].

A key issue in NK cell-based immunotherapy is their relatively short lifespan post-infusion, necessitating strategies for enhancing persistence. The introduction of cytokine preconditioning, feeder cell co-culture, and genetic modifications to improve metabolic fitness has demonstrated some success in prolonging NK cell activity in vivo [8,9]. Additionally, the combination of CAR-NK cells with checkpoint blockade inhibitors, such as anti-*PD-1* or anti-*TIGIT*, has been shown to restore NK cell cytotoxicity in immunosuppressive tumor settings [10,11].

Another area of interest is the modulation of the tumor microenvironment to improve NK cell infiltration and function. Hypoxia and the presence of immunosuppressive cells such as regulatory T cells and myeloid-derived suppressor cells create barriers to effective NK cell activity.

Strategies targeting metabolic pathways, such as glycolytic enhancement or modulation of mitochondrial function, have been explored to counteract these effects [16,17].

NK cell-based immunotherapy has emerged as a powerful tool in cancer treatment, offering advantages over traditional T cell-based approaches, particularly in terms of safety and off-the-shelf availability.

However, tumor-induced immunosuppression and the limited persistence of NK cells remain significant hurdles.

Advances in CAR-NK engineering, cytokine preconditioning, and combination therapies with immune checkpoint inhibitors hold promise for overcoming these challenges.

To maximize clinical efficacy, future research should prioritize strategies to enhance NK cell persistence, improve tumor infiltration, and modulate the immunosuppressive tumor microenvironment.

Integrating NK cell therapies with personalized medicine approaches may further refine their application in diverse malignancies. Ultimately, addressing these challenges will be crucial in realizing the full potential of NK cell-based immunotherapies as a mainstream cancer treatment strategy. With continued scientific innovation and clinical refinement, NK cell-based therapies are poised not only to complement—but to redefine—the future landscape of cancer immunotherapy.

## Figures and Tables

**Figure 1 biomedicines-13-00857-f001:**
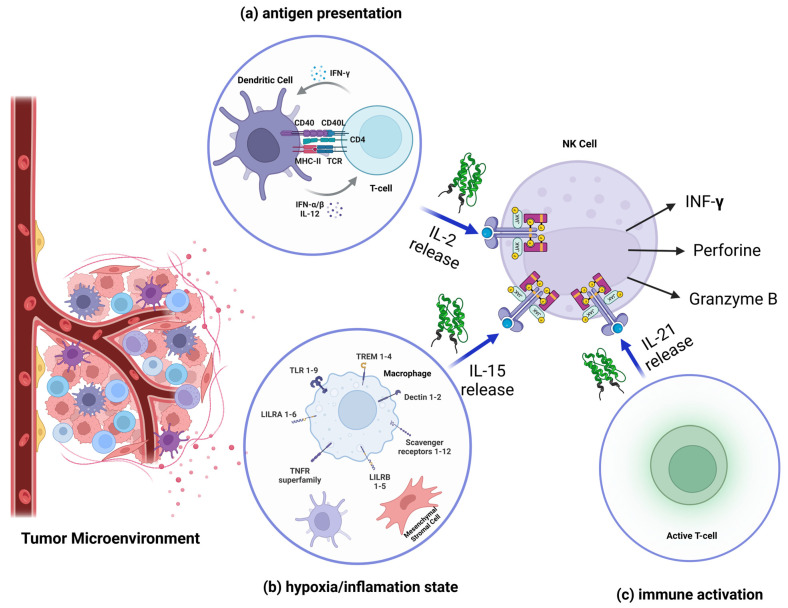
Summary of natural killer cell activation in a tumor environment. Created with BioRender.com. On the left side of the figure, a general representation of the tumor environment can be observed. The infiltration of various cells, including those of the immune system, is visible. Dendritic cells can activate T lymphocytes by presenting tumor antigens, which in turn release IL-2 and IL-21 (**a**–**c**). Hypoxia, DNA damage of inflammation, may occur, leading to the activation of macrophages, stromal cells, and dendritic cells, which release IL-15 (**b**). These three cytokines ultimately activate their receptors on the membrane of naïve natural killer cells, promoting their activation and proliferation.

**Figure 2 biomedicines-13-00857-f002:**
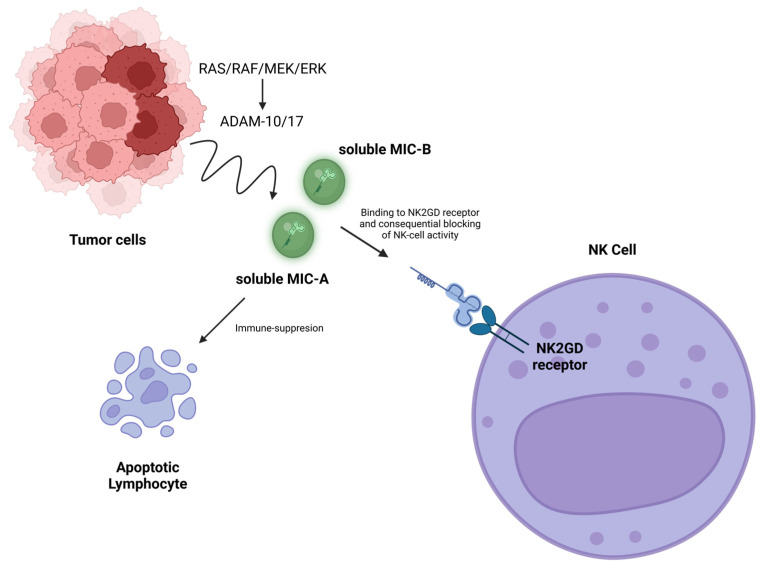
Mechanism of action of MIC-A and MIC-B. Natural killer (NK) cells have a receptor on their surface called NKG2D. Created with BioRender.com. Transformed or infected cells, in turn, express MIC-A and MIC-B molecules on their surface. Under physiological conditions, NK cells recognize and bind to transformed or infected cells through the interaction between MIC-A/B and the NKG2D receptor, triggering a cytotoxic response against them. However, tumors have developed complex mechanisms NS through pathways such as RAS/RAF/MEK/ERK, MIC-A/B molecules are cleaved from the membrane by metalloproteases (ADAM-10/17), becoming soluble. This process blocks NK cell receptors, suppressing their activity, and also attracts cells that suppress immune activity within the tumor microenvironment.

**Figure 3 biomedicines-13-00857-f003:**
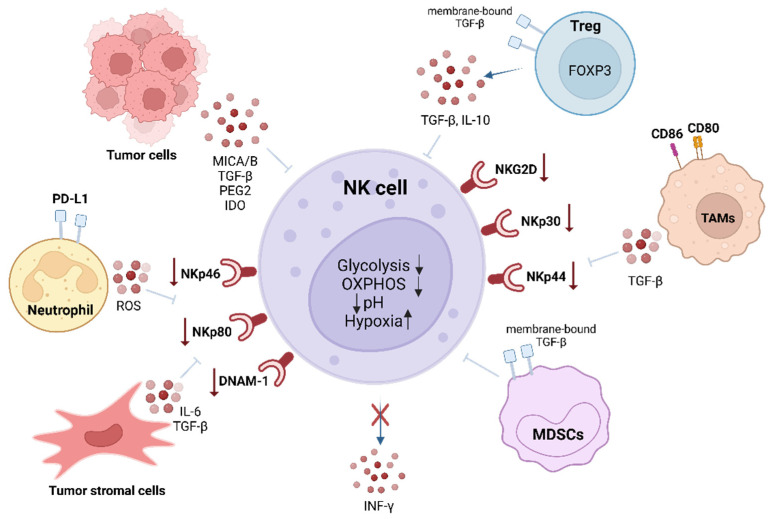
Immunosuppressive interactions between NK cells and the tumor microenvironment. Created with BioRender.com. The figure illustrates the complex network of inhibitory signals within the tumor microenvironment that impair NK cell function. Tumor cells, stromal cells, and immunosuppressive immune cells—including regulatory T cells (Tregs), myeloid-derived suppressor cells (MDSCs), tumor-associated macrophages (TAMs), and neutrophils—release cytokines and express surface molecules that downregulate NK cell activation, thereby reducing their antitumor response. Different arrow types indicate various mechanisms of suppression, including soluble factor release (red arrows) and receptor-ligand interactions (blue arrows).

**Figure 4 biomedicines-13-00857-f004:**
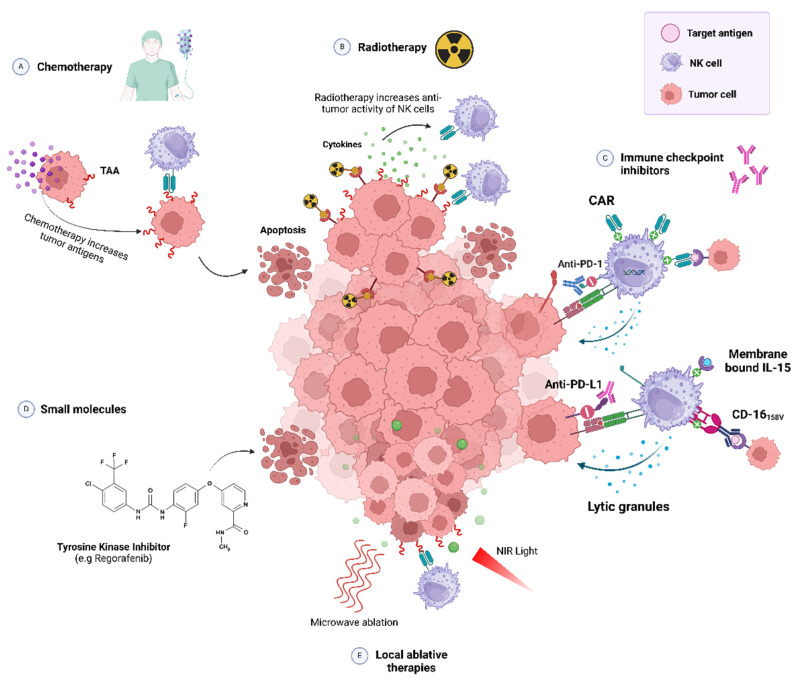
Possible combination therapies with NK cells. Created with BioRender.com. (**A**) Combination with chemotherapy. Low-dose chemotherapy has an immunomodulatory effect that increases tumor-associated antigens (TAAs) on tumor cells, which facilitates their recognition by the chimeric antigen receptor (CAR) on NK cells. (**B**) Combination with radiotherapy. Radiotherapy induces the release of different types of cytokines by the tumor cells and increases the presence of TAAs by enhancing the anti-tumor activity of CAR-NK cells. (**C**) Combination with immune checkpoint inhibitors. Combination with monoclonal antibodies potentiates NK cell activity against tumors by antibody-dependent cell-mediated cytotoxicity via high affinity non-cleavable hnCD16 (158 V) modified on the surface of NK cells or immune checkpoint inhibitors that block programmed cell death protein 1 (PD-1) on the surface of NK cells or inhibitory ligands on tumor cell membranes such as programmed cell death ligand 1 (PD-L1). In turn, NK cell activity can be stimulated in vivo by expression of the IL-15 receptor fusion protein and specifically targeting antigens on tumor cells by CAR. (**D**) Combination with small molecules. Molecular inhibitors disrupt signal transduction pathways through different mechanisms of action. Some are protein kinase inhibitors (e.g. Regorafenib), which interrupt protein kinase transduction signaling pathways, leading to cell apoptosis. In combination with NK cells, a synergistic effect is observed compared to monotherapies. (**E**) Combination with local ablative therapies. Local tumor ablative therapies such as photothermal therapies or microwave ablation release immunomodulatory factors such as tumor antigens and cytokines that enhance the anti-tumor response of immune cells; therefore, combining them with NK cells may enhance their therapeutic effect.

**Table 1 biomedicines-13-00857-t001:** Summary of natural killer (NK) cell receptors classified as activating or inhibitory, including their main ligands, associated signaling pathways, and functional roles in the context of tumor immunosurveillance.

Receptor	Type	Ligand	Signaling Pathway	Function
NKG2D	Activating	MICA/B, ULBPs	DAP10	Cytotoxicity, cytokine production
NKp30	Activating	B7-H6	ITAM	Tumor cell recognition
NKp44	Activating	Unknown/Stress ligands	ITAM	Enhances NK activation
NKp46	Activating	Viral hemagglutinins	ITAM	Viral defense
CD226	Activating	CD155 (PVR)	ITAM	Co-stimulation of NK response
CD16	Activating	Fc region of antibodies	FcγRIIIa (CD16)	ADCC
NKG2A	Inhibitory	HLA-E	ITIM	Inhibition of NK activity
KIR	Inhibitory	HLA-A/B/C	ITIM	Self-tolerance, inhibition
TIGIT	Inhibitory	CD155 (PVR)	ITIM	Suppresses NK function
CD94/NKG2A	Inhibitory	HLA-E	ITIM	Inhibits cytotoxicity
PD-1	Inhibitory	PD-L1	ITIM	Suppresses NK response
CD96	Inhibitory	CD155	ITIM	Inhibits IFN-γ production
TIM-3	Inhibitory	Galectin-9	ITIM	Induces exhaustion

**Table 2 biomedicines-13-00857-t002:** Current clinical trials investigating CAR-NK in cancer immunotherapy, now progressing beyond phase 1.

CAR Target	NK Cell Source	Targeting Tumor	National Clinical Trial Identifier
CD19	Umbilical cord blood (UCB)	Hematological malignancies	NCT03056339
	Nonreferred	B cell hematologic malignancies	NCT05570188
	Hematopoietic progenitor cells (HPCs)	B-cell lymphoma	NCT05654038
CD70	Umbilical cord blood (UCB)	Hematological malignancies	NCT05092451
	Umbilical cord blood (UCB)	Solid tumors	NCT05703854
CD19/CD70	Umbilical cord blood (UCB)	B-cell non-Hodgkin lymphoma	NCT05842707
CD19/CD28	Umbilical cord blood (UCB)	B-cell non-Hodgkin lymphoma	NCT03579927
CD5	Umbilical cord blood (UCB)	Hematological malignancies	NCT05110742
CD7	Peripheral blood mononuclear cells (PBMCs)	Leukemia and lymphoma	NCT02742727
CD123	Peripheral blood mononuclear cells (PBMCs)	Acute myeloid leukemia and blastic plasmacytoid dendritic cell neoplasm	NCT06006403
PD-L1	NK92	Gastroesophageal junction cancers or head and neck squamous cell carcinoma	NCT04847466
Cladin6	Peripheral blood mononuclear cells (PBMCs)	Reproductive system tumors	NCT05410717
BCMA	NK92	Multiple myeloma	NCT03940833
CD33	NK92	Acute myeloid leukemia	NCT02944162
MUC1	Peripheral blood mononuclear cells (PBMCs)	Solid tumors	NCT02839954
Robo1	NK92	Pancreatic cancer	NCT03941457
TROP2	Umbilical cord blood (UCB)	Ovarian cancer, mesonephric-like adenocarcinoma, and pancreatic cancer	NCT05922930

**Table 3 biomedicines-13-00857-t003:** Comparison of main characteristics and disadvantages of NK cell therapies, CAR-T cell therapies, and CAR-NK therapies.

Characteristic/Therapy	NK Cells	CAR-T Cells	CAR-NK Cells
Mechanism of action	Recognition and direct lysis of tumor cells	CAR receptor-mediated antigen recognition and cell lysis	CAR receptor-mediated antigen recognition and cell lysis
Activation time	Fast, can act immediately after infusion	Requires weeks due to necessary ex vivo expansion and activation	Requires ex vivo expansion and activation, like CAR-T cells
Shelf life	Short, limited to days or weeks	Prolonged, can last for months or even years	Variable, although shorter than that of CAR-T cells
Side effects	Minor compared to CAR-T	Significant, including cytokine release syndrome (CRS) and neurotoxicity	Minor compared to CAR-T, although there is risk of CRS
Applicability	Hematological and some solid tumors	Leukemias and lymphomas, with limited efficacy in solid tumors	Promising for solid and hematologic tumors (in development)
Response rate	Good, although it varies according to the type of tumor	High in certain types of leukemias and lymphomas	Promising, although still in development
Disadvantages	Short shelf life and resistance of the tumor microenvironment	High cost and complexity in production and handling	Need for optimization in production and efficacy

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
