# Peer review of "The Role of NK Cells in Cancer Immunotherapy: Mechanisms, Evasion Strategies, and Therapeutic Advances"

_biomedicines, 2025, doi:10.3390/biomedicines13040857_

Round 1

Reviewer 1 Report

Comments and Suggestions for Authors

Authors are commended for compiling a decade of literature on NK cells and immunotherapy covering  mechanisms, evasion strategies and therapeutic advances.

However minor revisions are required.

  1. Fig 1. Did the author mean 'left' side of figure, a general representation of the tumor environment. Also label the cells types in fig 1 as done in fig 3/4.
  2. Fig 2. is it an apoptotic cell? Please label.

Author Response

Authors are commended for compiling a decade of literature on NK cells and immunotherapy covering mechanisms, evasion strategies and therapeutic advances.

However minor revisions are required.

  1. Fig 1. Did the author mean 'left' side of figure, a general representation of the tumor environment. Also label the cells types in fig 1 as done in fig 3/4.

As Reviewer 1 suggests, the cells types in fig 1 has been labeled.

  1. Fig 2. is it an apoptotic cell? Please label.

Indeed, it is a cell. To avoid confusion, it has been labeled.

Reviewer 2 Report

Comments and Suggestions for Authors

Comments:

  1. The abstract writing does not fully comply with the standard format for review articles. Please revise.

  2. The overall structure of the article does not align well with the typical framework of a review article, especially with sections like "Introduction" and "Results." We suggest restructuring.

  3. The author should include in the introduction section other published reviews on NK cells in cancer therapy, describe which issues still need clarification or attention, and highlight the focus of this review.

  4. Line 180: What is ITAMs?

  5. Line 273: The heading 2.1.3.1 appears redundant, as there is no 2.1.3.2 in the content. We suggest removing it. The author has mentioned these interleukin family factors earlier, so starting a new paragraph here would not seem abrupt.

  6. Some specialized abbreviations are redundantly defined with their full terms throughout the manuscript. Please review the entire manuscript to avoid unnecessary repetition, such as TME. However, some abbreviations lack their full terms, such as TAM.

  7. The Discussion section seems to add little value and repeats content from the introduction. If the author intends to summarize the entire article, the Discussion and Conclusions sections could be merged under a single heading "Summary and Future Perspectives."

  8. Overall, this review provides a comprehensive summary of NK cell-related cytokines and regulatory signals. The content is rich, well-evidenced, and represents a valuable reference for learning and study in this field.

Author Response

Comments:

  1. The abstract writing does not fully comply with the standard format for review articles. Please revise.

We thank the Reviewer for this helpful observation. In response, we have rewritten the abstract following the structured format typically used in review articles, including clear subsections: Background/Objectives, Methods, Results, and Conclusions. The new version is now limited to 200 words, as per Biomedicines guidelines. This revised abstract provides a concise summary of the review’s scope, methodology, main findings, and future perspectives. The updated abstract can be found on page 1 and is highlighted in blue.

  1. The overall structure of the article does not align well with the typical framework of a review article, especially with sections like "Introduction" and "Results." We suggest restructuring.

We appreciate the Reviewer’s suggestion regarding the structure of the manuscript. In response, we have restructured the article to better align with the standard format of a review article. The former “Results” section has been renamed to “Mechanisms of NK Cell Activity in Cancer”, which more accurately reflects its descriptive and integrative nature.

We have also reorganized and clarified the section and subsection headings to improve the manuscript’s logical flow and thematic coherence. Specifically, we have revised the structure to include the following major sections:

  1. Introduction
  2. Mechanisms of NK cell activity in cancer
  3. Tumor evasion mechanisms
  4. NK Cell-Based Immunotherapies
  5. Challenges and future directions
  6. Discussion
  7. Conclusions

These structural changes aim to enhance the clarity and narrative consistency expected in a review article. All modifications have been highlighted in green in the revised manuscript for easy identification.

  1. The author should include in the introduction section other published reviews on NK cells in cancer therapy, describe which issues still need clarification or attention, and highlight the focus of this review.

We have revised the Introduction section to include references to several recent and relevant reviews on NK cells in cancer immunotherapy. These additions help contextualize the current state of the field and position our review in relation to previous work.

We have also explicitly addressed the knowledge gaps that still require clarification, such as challenges related to NK cell persistence, expansion, tumor immune evasion, and their application in solid tumors. Furthermore, we have clearly defined the specific focus of our review, emphasizing how it contributes to advancing the field by integrating mechanistic insights and highlighting innovative therapeutic strategies such as CAR-NK cells and gene editing technologies.

All changes have been marked in pink in the revised manuscript for ease of identification.

  1. Line 180: What is ITAMs?

We thank the Reviewer for pointing this out. We have now clarified the term in the text by introducing the full name Immunoreceptor Tyrosine-based Activation Motifs (ITAMs) on first mention.

This addition has been marked in grey in the revised manuscript.

  1. Line 273: The heading 2.1.3.1 appears redundant, as there is no 2.1.3.2 in the content. We suggest removing it. The author has mentioned these interleukin family factors earlier, so starting a new paragraph here would not seem abrupt.

We thank the Reviewer for this helpful observation. Following the Reviewer’s recommendation, we have removed the heading 2.1.3.1 to avoid redundancy, as there was no corresponding 2.1.3.2. Additionally, we have reorganized the section so that the discussion of interleukin family factors flows naturally in the subsequent paragraph, without the need for a separate heading.

  1. Some specialized abbreviations are redundantly defined with their full terms throughout the manuscript. Please review the entire manuscript to avoid unnecessary repetition, such as TME. However, some abbreviations lack their full terms, such as TAM.

We appreciate the Reviewer’s observation. We have carefully reviewed the manuscript to eliminate repeated definitions of abbreviations already introduced, such as TME and CAR-NK cells, to improve clarity and conciseness. Additionally, we have ensured that all abbreviations—including TAM (Tumor-Associated Macrophages), MDSC, ADCC, and CIML NK cells—are clearly defined at their first occurrence in the text.

These revisions have been marked in clear yellow in the updated manuscript.

  1. The Discussion section seems to add little value and repeats content from the introduction. If the author intends to summarize the entire article, the Discussion and Conclusions sections could be merged under a single heading "Summary and Future Perspectives."

In response, we have merged the original Discussion and Conclusions sections into a single, comprehensive section entitled “Summary and Future Perspectives.” This revised section avoids repetition and provides a more integrated synthesis of the key insights presented throughout the review.

The updated version summarizes current findings, discusses persistent challenges—such as immune evasion and limited NK cell persistence—and outlines future directions with emphasis on metabolic modulation, genetic engineering, and combinatorial approaches.

These modifications have been marked in dark green in the revised manuscript for easy identification.

Overall, this review provides a comprehensive summary of NK cell-related cytokines and regulatory signals. The content is rich, well-evidenced, and represents a valuable reference for learning and study in this field.

We appreciate the valuable discernment of Reviewer 2 to our work. We greatly value the recognition of the comprehensive summary and the quality of the content in our review. We are glad to hear that the article provides valuable insight into NK cell-related cytokines and regulatory signals. We will continue to refine and improve the manuscript to ensure it serves as a valuable reference for those studying this field.

We hope that the revised version of our manuscript meets the expectations of the reviewers and the editorial team. We sincerely thank you for the constructive feedback, which has greatly contributed to improving the clarity, coherence, and overall quality of our review. We remain at your disposal for any further modifications that may be required and look forward to your favorable consideration.

Reviewer 3 Report

Comments and Suggestions for Authors

This review discusses the role of NK cells in tumor surveillance, focusing on their activation, inhibition, and tumor evasion mechanisms. It highlights NK cell-based immunotherapies, including CAR-NK cells and adoptive transfer, emphasizing their safety advantages over CAR-T therapies. Despite promising results, challenges such as limited persistence and tumor-induced immunosuppression remain. Addressing these issues is crucial for optimizing NK cell therapies and advancing next-generation immunotherapeutics.

The topic of the review is very intriguing and emerging.

The review is well-written, the Figures are well-performed, and the tables are very useful.

However, I have only some suggestions.

-Some phrases could be rephrased to improve readability. The original text contains some redundancies and long-winded explanations. The authors could streamline the sentences while maintaining technical accuracy.  The ideas should be connected more smoothly. Some sentences are dense and could benefit from simplification or restructuring for better readability.

-The explanation of NK cell receptors is strong, but the authors could enhance transitions to improve coherence. In general, transitions between sections (e.g., between inhibitory receptors and activating receptors) could be smoother to improve coherence.

  • About refs and citations:  Ensure that key claims, especially novel ones, are well-supported with up-to-date references. Some references (e.g., [2,3]) are cited repeatedly across multiple concepts. Clarifying whether they provide direct evidence for each claim would improve rigor. Some new papers about the signaling axis involving MICA/B shedding in ADAM 10 are not cited. This shedding is linked to Androgen receptor activation in melanoma aggressiveness and tumor escape by NK cells through NKG2D.  Please improve. This also links the increase of MICA/B with poor prognosis of melanoma patients.

-The authors should consider adding citations where they discuss novel therapies.

-The conclusions should be more impactful.

-Minor grammatical adjustments should be made for smoother reading.

- "towards self-cells" , is better "toward self-cells" 

-"On the one hand it has been shown that..." This phrase needs a corresponding "on the other hand" for balance.

-The use of "inhibitory phenotype" could be defined more precisely.

-Ensure consistent formatting of receptor names

- A table or diagram summarizing NK cell inhibitory and activating pathways and receptors for visual clarity could be appreciated. 

Author Response

This review discusses the role of NK cells in tumor surveillance, focusing on their activation, inhibition, and tumor evasion mechanisms. It highlights NK cell-based immunotherapies, including CAR-NK cells and adoptive transfer, emphasizing their safety advantages over CAR-T therapies. Despite promising results, challenges such as limited persistence and tumor-induced immunosuppression remain. Addressing these issues is crucial for optimizing NK cell therapies and advancing next-generation immunotherapeutics.

The topic of the review is very intriguing and emerging.

The review is well-written, the Figures are well-performed, and the tables are very useful.

Thank you for the kind and encouraging feedback. We are pleased to hear the Reviewer 3 found the topic of the review intriguing and emerging, and that the manuscript is well-written, with the figures and tables being useful and well-executed. We value his/her positive assessment of the content and will continue to refine the manuscript to ensure clarity and completeness in presenting the role of NK cells in tumor surveillance and immunotherapy.

However, I have only some suggestions.

  1. Some phrases could be rephrased to improve readability. The original text contains some redundancies and long-winded explanations. The authors could streamline the sentences while maintaining technical accuracy.  The ideas should be connected more smoothly. Some sentences are dense and could benefit from simplification or restructuring for better readability.

We have carefully reviewed the manuscript and rephrased certain phrases to improve readability and eliminate redundancies. We also streamlined some of the longer explanations to make the content more concise without compromising technical accuracy. Additionally, we restructured sentences where needed to ensure a smoother flow of ideas and to enhance the overall readability of the manuscript.

These changes have been highlighted in yellow in the revised manuscript. 

- In line 139 we replaced” These inhibitory receptors are important because their activation inhibits the destruction of normal cells presenting these HLA molecules and prevents them from being attacked by NK cells” for “Crucially, the activation of these receptors inhibits the destruction of normal cells containing these HLA molecules, protecting them from NK cells”.

- In lines 492-495 we summarized the content in one sentence.

- In lines 680,681 we summarized the content.

- In lines 1140-1143 we summarized the content.

  1. The explanation of NK cell receptors is strong, but the authors could enhance transitions to improve coherence. In general, transitions between sections (e.g., between inhibitory receptors and activating receptors) could be smoother to improve coherence.

We appreciate the Reviewer’s suggestion. In response, we have introduced a clarifying transition between the sections discussing inhibitory and activating NK cell receptors. This addition improves the narrative flow and helps the reader follow the change in focus more clearly.

This revision has been marked in red in the revised manuscript.

  1. About refs and citations:  Ensure that key claims, especially novel ones, are well-supported with up-to-date references. Some references (e.g., [2,3]) are cited repeatedly across multiple concepts. Clarifying whether they provide direct evidence for each claim would improve rigor. Some new papers about the signaling axis involving MICA/B shedding in ADAM 10 are not cited. This shedding is linked to Androgen receptor activation in melanoma aggressiveness and tumor escape by NK cells through NKG2D.  Please improve. This also links the increase of MICA/B with poor prognosis of melanoma patients.

We agree that ensuring key claims, especially novel ones, are well-supported with up-to-date references is crucial for the manuscript’s rigor. In response to Reviewer´s suggestion, we have carefully reviewed the references and added additional citations where necessary to ensure that all claims, particularly those that are novel, are adequately supported by recent studies.

We also appreciate the suggestion regarding the signaling axis involving MICA/B shedding in ADAM10, particularly in the context of androgen receptor activation in melanoma aggressiveness and tumor escape by NK cells through NKG2D. We have now included relevant recent papers discussing this signaling pathway and the link between MICA/B increase and poor prognosis in melanoma patients to strengthen this aspect of the review.

  1. The authors should consider adding citations where they discuss novel therapies.

In response to the Reviewer's comment, we have incorporated these references in the clinical trials section to better contextualize the information presented in our manuscript.

While these references could have been placed in another section of the article, we believe that including them in the clinical trials section allows for better integration with the discussion on the impact of these strategies in clinical practice. Furthermore, other novel therapies have also been evaluated, and we have added the corresponding citations to support these statements. This includes references to recent studies and clinical trials that highlight the latest advancements. We believe this strengthens the manuscript and ensures that the discussion is well-supported by up-to-date evidence. All changes have been marked in navy blue in the revised manuscript.

  1. The conclusions should be more impactful.

We appreciate the Reviewer’s suggestion. In fact, a similar recommendation was made by another reviewer, who advised us to restructure the final sections of the manuscript. In response, we have integrated the Discussion and Conclusions into a single, more cohesive and impactful section entitled “Summary and Future Perspectives.”

In addition, we have revised the content of this section to reinforce the main takeaways of the review in a more direct and compelling manner, clearly emphasizing the relevance of current challenges and the potential of future therapeutic strategies.

  1. Minor grammatical adjustments should be made for smoother reading.

We have carefully reviewed the manuscript for minor grammatical adjustments and have made the necessary corrections to improve the flow and readability of the text. These changes help to ensure smoother reading while maintaining the technical accuracy of the content.

  1. "towards self-cells", is better "toward self-cells" 

We thank the Reviewer for this observation. We have revised the phrase to use “toward” instead of “towards”, in accordance with standard American English conventions and for consistency throughout the manuscript.

  1. "On the one hand it has been shown that..." This phrase needs a corresponding "on the other hand" for balance.

We thank the Reviewer for this stylistic observation. In response, we have revised both relevant sections of the manuscript to ensure proper parallel structure. Specifically, we have added the corresponding “on the other hand” to provide a balanced contrast with “on the one hand”, improving rhetorical symmetry and clarity in both passages.

This correction has been marked in blue in the revised manuscript.

  1. The use of "inhibitory phenotype" could be defined more precisely.

We agree with the Reviewer that the term “inhibitory phenotype” could benefit from further clarification. In response, we have revised the relevant paragraph to include a precise definition of this concept, describing how it is characterized by the upregulation of inhibitory receptors (such as NKG2A, TIGIT, or PD-1) and the suppression of NK cell cytotoxicity and cytokine production under the influence of the tumor microenvironment. This clarification helps contextualize its functional relevance in NK cell regulation and tumor immune evasion.

This addition has been marked in navy blue font in the revised manuscript.

  1. Ensure consistent formatting of receptor names

We thank the Reviewer for these helpful observations. In response, we have added the full term “T cell immunoreceptor with Ig and ITIM domains” at the first mention of TIGIT in the Introduction. The revised sentence now reads:

“Additionally, research by Harjunpää and Guillerey on the checkpoint receptor T cell immunoreceptor with Ig and ITIM domains (TIGIT), as well as studies on the interaction of nectin-1 with CD96, will be examined to highlight critical regulatory pathways.”

We have also reviewed the manuscript to ensure consistent formatting of receptor names such as PD-1, TIGIT, NKp30, and TIM-3. All instances within the main text have been standardized according to current nomenclature conventions.

The lowercase form “Tim-3” appears only within the title of a cited reference and has been left unchanged to preserve the original publication title.

This change has been marked in orange in the revised manuscript.

  1. A table or diagram summarizing NK cell inhibitory and activating pathways and receptors for visual clarity could be appreciated. 

we have included Table 1, which provides a concise visual summary of the main NK cell receptors. The table classifies these receptors as activating or inhibitory, and includes their respective ligands, signaling pathways, and functional roles in tumor immunosurveillance.

This table has been inserted within section 2.1.1 “Recognition of tumor cells by inhibitory and activating receptors on NK cells”, immediately after the paragraph discussing CD96, DNAM-1, and TIGIT, and before the detailed description of activating receptors. This placement ensures logical continuity and enhances the clarity of the manuscript.

We hope that the revised version of our manuscript meets the expectations of the reviewers and the editorial team. We sincerely thank you for the constructive feedback, which has greatly contributed to improving the clarity, coherence, and overall quality of our review. We remain at your disposal for any further modifications that may be required and look forward to your favorable consideration.